# FERD: Fairness-Enhanced Data-Free Adversarial Robustness Distillation

**Zhengxiao Li**[1*]**, Liming Lu**[1*]**, Xu Zheng**[2,3]**, Siyuan Liang**[4]**, Zhenghan Chen**[5†]**,
Yongbin Zhou**[1]**, Shuchao Pang**[1†]

[1]Nanjing University of Science and Technology [2]HKUST(GZ)
[3]INSAIT, Sofia University, St. Kliment Ohridski [4]Nanyang Technological University
[5]STCA, Microsoft
{lizhengxiao, luliming}@njust.edu.cn

## ABSTRACT

Data-Free Robustness Distillation (DFRD) aims to transfer the robustness from the teacher to the student without accessing the training data. While existing methods focus on overall robustness, they overlook the robust fairness issues, leading to severe disparity of robustness across different categories. In this paper, we find two key problems: (1) student model distilled with equal class proportion data behaves significantly differently across distinct categories; and (2) the robustness of student model is not stable across different attacks targets. To bridge these gaps, we present the first Fairness-Enhanced data-free adversarial Robustness Distillation (FERD) framework to adjust the proportion and distribution of adversarial examples. For the proportion, FERD adopts a robustness-guided class reweighting strategy to synthesize more samples for the less robust categories, thereby improving robustness of them. For the distribution, FERD generates complementary data samples for advanced robustness distillation. It generates Fairness-Aware Examples (FAEs) by enforcing a uniformity constraint on feature-level predictions, which suppress the dominance of class-specific non-robust features, providing a more balanced representation across all categories. Then, FERD constructs Uniform-Target Adversarial Examples (UTAEs) from FAEs by applying a uniform target class constraint to avoid biased attack directions, which distributes the attack targets across all categories and prevents overfitting to specific vulnerable categories. Extensive experiments on three public datasets demonstrate that FERD achieves state-of-the-art worst-class robustness and NSD under all adversarial attacks. For instance, FERD improves worst-class robustness by up to 11.3% and reduces NSD by 0.077 compared to the optimal baseline on CIFAR-10 with MobileNet-V2. Our code is available at: https://github.com/mayaobuduyao/FERD.

## 1 INTRODUCTION

With the widespread use of Deep Neural Networks (DNNs) (Goswami et al., 2018; Gongye et al., 2024; Lim et al., 2024), the deployment of lightweight models on edge devices has become increasingly important (Zhu et al., 2025; Pang et al., 2025; Liu et al., 2024). However, a large number of studies have shown that these lightweight models are weakly robust in the face of adversarial attacks (Ma et al., 2021; Li et al., 2022; Croce & Hein, 2020; Bai et al., 2023). Traditional adversarial training methods (Jia et al., 2022; Hsiung et al., 2023; Jia et al., 2024b;a), though showing significant advantages on large models, are difficult to achieve desirable results in lightweight models (Xia et al., 2025; Wang et al., 2024b; Ye et al., 2019; Huang et al., 2021). To enhance the defense capability of lightweight models, researchers have proposed the concept of adversarial robustness distillation (Zi et al., 2021; Zhu et al., 2021; Huang et al., 2023; Yue et al., 2024; Zhu et al., 2023;

---

[*]Equal contribution.
[†]Corresponding authors.

Lu et al., 2025; Gu et al., 2025), which aims to migrate the defense capability of the robust teacher to the student, thereby enhancing the latter's performance in an adversarial setting.

However, in practice, raw training data for the teacher model are often unavailable, making it difficult to apply traditional distillation methods directly. To break through this limitation, researchers propose the Data-Free Knowledge Distillation (DFKD) method (Micaelli & Storkey, 2019; Fang et al., 2021; Yin et al., 2020; Fang et al., 2022), which synthesizes alternative samples through a training generator to simulate the original data distribution and achieve knowledge transfer. Based on this, the Data-Free Robustness Distillation (DFRD) method (Yuan et al., 2024; Wang et al., 2024a; Zhou et al., 2024) is further developed to combine the generation and distillation mechanisms to effectively deliver robustness without the need for real data, providing a novel solution for resource-constrained and data-unavailable environments for model defense.

Despite their effectiveness, existing DFRD methods mainly aim to enhance the overall robustness of the student, while overlooking a critical issue: robust fairness (Sun et al., 2023; Yue et al., 2023; Zhao et al., 2024). The robust model may exhibit strong resistance to adversarial attacks on specific categories and remain vulnerable in others, leading to inconsistent robustness performance across different categories. It impacts the reliability and fairness of the model in practical applications.

In this paper, we make the first attempt at investigating the robust fairness problem in the context of DFRD. We find that although the student tends to inherit the teacher's class-wise robustness pattern, the inter-class robustness gap is significantly amplified in the distillation process. We find two phenomena affecting robust fairness: (1) students distilled with equally distributed synthetic data still show class-wise robustness discrepancies; and (2) the success rate of adversarial attacks on students varies significantly depending on the target class. Based on these findings, we propose a Fairness-Enhanced data-free adversarial Robustness Distillation (FERD) framework by adjusting the proportion and distribution of the synthetic samples to mitigate these problems.

Specifically, for the proportion, we introduce a robustness-guided class reweighting strategy that encourages the generator to synthesize more samples from weakly robust categories, thereby compensating for their vulnerability and promoting fairness in robustness. For the distribution, we propose to generate complementary data samples for advanced robustness and fairness distillation. Firstly, we generate Fairness-Aware Examples (FAEs) by enforcing a uniformity constraint on feature-level predictions that are closely associated with non-robust representations. This helps suppress the dominance of class-specific non-robust features, ensuring that the benign samples provide a more balanced representation across all classes. To avoid biased attack directions, we further construct Uniform-Target Adversarial Examples (UTAEs) from FAEs by applying a uniform target class constraint during adversarial generation. It distributes the attack targets evenly across all categories and prevents overfitting to specific vulnerable categories.

Experiments on the three datasets (CIFAR-10, CIFAR-100, and Tiny-ImageNet) show that compared with baseline methods, FERD improves 11.3%, 5.7%, 6.2% and 6.6% on the worst-class robustness against FGSM (Goodfellow et al., 2014), PGD-20 (Madry et al., 2017), $CW_\infty$ (Carlini & Wagner, 2017), and AutoAttack (AA) (Croce & Hein, 2020), respectively, alleviating the robustness bias problem in the DFRD to a certain extent.

Our contributions can be summarized as follows: (I) We make the first attempt at investigating the problem of robust fairness in DFRD, revealing that uniform distribution among original data categories and attack target bias are the two key factors affecting fairness. (II) We propose the FERD framework, which enhances robust fairness at both the proportional and distributional levels through robustness-guided category reweighting and distribution-aware sample generation mechanisms. (III) Experiments have demonstrated that FERD significantly enhances the robustness and fairness of the student model, and the robust accuracy in the weakest class has been improved by 11.3% compared with the existing optimal methods, effectively alleviating the robust bias phenomenon in DFRD.

## 2 RELATED WORK

### 2.1 DATA-FREE ROBUSTNESS DISTILLATION

DFRD aims to transfer the robustness from teacher to student using synthetic samples instead of teacher's original training data. DFARD (Wang et al., 2024a) first defines the concept of DFRD and

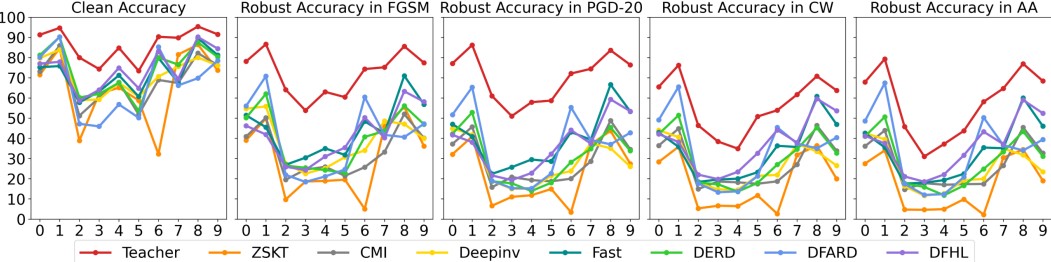

Figure 1: Comparison of the accuracy of the teacher and the students distilled from different DFRD methods under benign and adversarial examples. The blue line represents the teacher and the other lines correspond to the students. The horizontal axis indicates the category, and the vertical axis indicates the accuracy.

proves that the difficulty lies in the lower upper bound of knowledge transfer information. They propose an interactive temperature adjustment strategy and an adaptive generator to solve the problem. DERD (Zhou et al., 2024) takes a homogenized expert guidance strategy. Both clean and robust knowledge are distilled from clean and robust teachers respectively, using the same synthetic data. To coordinate the gradients of the clean and robust distillation tasks, DERD also introduces a stochastic gradient aggregation module, thereby optimizing the trade-off between robustness and accuracy. DFHL (Yuan et al., 2024) proposes the concept of High-Entropy Examples (HEEs), which can characterize a more complete shape of the classification boundary. Distillation on HEEs achieves the best balance between clean accuracy and robustness. It is worth mentioning that although DFKD does not involve robustness, we can transform it into DFRD by adding adversarial noise to synthetic samples during its distillation stage. For example, Fast (Fang et al., 2022) proposes a fast DFKD, which reuses the common features in the training data to synthesize different data instances. By generating adversarial examples on synthetic samples, a distilled robust student can be obtained. In this paper, we make the first attempt at solving the problem of robust fairness transfer in DFRD.

## 2.2 FAIRNESS IN ROBUSTNESS

While enhancing overall model robustness is a common goal, it inevitably leads to significant class-wise performance discrepancies: models become highly robust for some categories while others remain vulnerable to adversarial attacks. FRL (Xu et al., 2021) is a pioneer work in highlighting this issue and introduces the concept of "robust fairness" to assess such class-wise robustness disparities. To address this issue, FRL proposes fairness constraints, adjusting decision margins and sample weights when these constraints are violated. BAT (Sun et al., 2023) identify distinct "source-class" and "target-class" unfairness within adversarial training, tackling these by adjusting per-class attack intensities and applying a uniform distribution constraint. Further methods include those by Fair-ARD (Yue et al., 2023), who improves student model robust fairness by increasing the weights of difficult classes, and ABSLD (Zhao et al., 2024), who focuses on adaptively reducing inter-class error risk gaps by modulating the class-wise smoothness of samples' soft labels during training. In this paper, we introduce FERD to simultaneously enhance model robustness and alleviate robust unfairness problems.

## 3 OBSERVATION OF FAIRNESS IN DFRD

In this section, we investigate whether robust fairness of the teacher is transferred to the student distilled from DFRD methods. We research the robust fairness performance of the models and defense effects against adversarial attacks from different target categories.

### 3.1 ROBUST FAIRNESS OF THE STUDENT

To evaluate the robust fairness of the student, we compare the classification performance of the teacher and the students distilled by five methods (ZSKT (Micaelli & Storkey, 2019), CMI (Fang et al., 2021), DeepInv (Yin et al., 2020), Fast (Fang et al., 2022), and DFHL (Yuan et al., 2024)) on

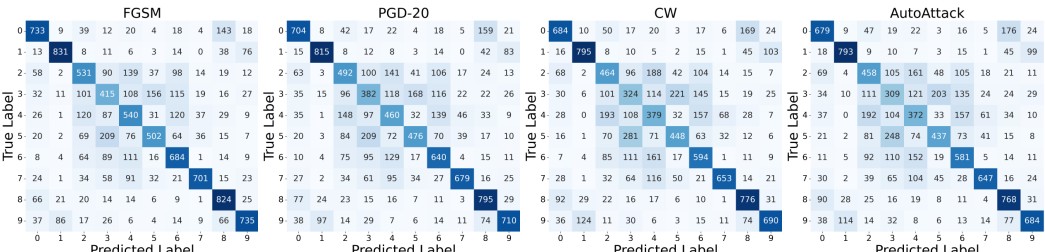

Figure 2: Confusion matrices under different adversarial attacks. The horizontal axis denotes predicted labels, and the vertical axis denotes true labels. Darker colors indicate a higher number of samples predicted as the correct class.

different classes. We measure the accuracy of each category under benign samples and four adversarial attacks (FGSM, PGD-20, $CW_\infty$, and AA). The results in Fig.1 show that the performance of the student on different categories follows a consistent trend with that of the teacher. Note that the sample labels all adopt a uniform sampling strategy in the aforementioned DFRD methods, in which case the student conducts robust distillation on equally distributed synthetic data. However, the robustness among different categories varies significantly. Therefore, we argue that sample number imbalance is the key to achieving class balance, because categories behave differently in robust distillation. Some categories are more difficult to achieve robustness, while others quickly reach a high level of robustness. We conjecture that by adjusting the weight of class sampling and increasing the number of samples from weakly robust classes, the problem of robust unfairness can be effectively alleviated. The proof of this conjecture is shown in Appendix A.1.

## 3.2 TARGET FAIRNESS OF ADVERSARIAL EXAMPLES

We make confusion matrices to visualize the classification results, as shown in Fig.2. We observe that the students' robustness against adversarial attacks on different targets varies. For samples with the original label of class 0, when the attack target is class 9, the student is more prone to misclassification, showing poor robustness; while for adversarial attacks on other classes, its robustness is relatively strong. This indicates that the student exhibits significant differences in defending against adversarial attacks on different targets and is more vulnerable to specific class adversarial attacks. We conjecture that robust distillation on adversarial examples with uniformly distributed targets can defend against attacks from different targets. The proof of this conjecture is shown in Appendix A.2.

## 4 METHODOLOGY

### 4.1 OVERALL FRAMEWORK

In the above section, we find two factors affecting robust fairness: (1) the student distilled on data with equal class proportions shows class-wise robustness discrepancies; and (2) the success rate of adversarial attacks on the student varies significantly depending on the target class.

In this section, we introduce our FERD framework and the overall framework is in Fig.3. The algorithm is shown in the Appendix B. For the first proportion problem, we introduce a robustness-guided class reweighting strategy that encourages the generator to synthesize more samples from weakly robust classes. For the second distribution problem, we impose uniform constraints on the feature space of synthetic samples and the generation direction of adversarial examples, respectively, making the targets of adversarial examples more uniformly distributed. Specifically, we firstly generate Fairness-Aware Examples(FAEs) by enforcing constraint on predictions from non-robust feature, which are highly related to adversarial targets. Then, to avoid biased attack directions, we further construct Uniform-Target Adversarial Examples (UTAEs) from FAEs by applying constraint during adversarial generation, preventing attacks focus on "vulnerable category".

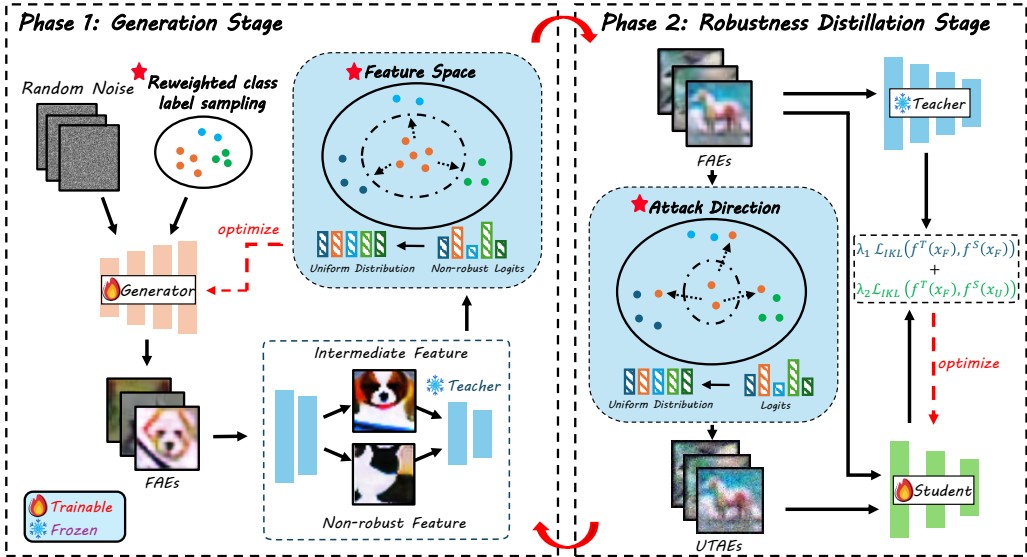

Figure 3: Framework of our FERD. In the generation stage, we use a robustness-guided class reweighting strategy to synthesize more weak-class samples and apply a uniformity constraint to non-robust feature predictions to generate FAEs. During robust distillation, we construct UTAEs from FAEs, using them respectively as benign samples and adversarial examples.

## 4.2 Class-Aware Sample Reweighting for Fair Distillation

In traditional DFRD, the labels $y_i$ of synthetic samples $x_i$ are sampled from a uniform distribution, as $y_i \sim \mathcal{U}(0, C-1)$, where $\mathcal{U}$ means uniform distribution and $C$ means class number. However, robust distillation on equally distributed data infers a significant robustness unfairness problem.

To mitigate this problem, we introduce the robustness-guided class reweighting strategy based on adversarial margin, aiming to guide the generator to synthesize more samples of categories with poorer robustness. Specifically , we first generate adversarial examples $x_i^{adv}$ from the synthetic samples $x_i$ using PGD-20 attack. Then, we calculate the adversarial margin $m_i$ of each adversarial sample $x_i^{adv}$ under the teacher model $f^T$:

$$m_i = \left(f^T(x_i^{adv})\right)_{y_i} - \max_{j \neq y_i} \left(f^T(x_i^{adv})\right)_j . \tag{1}$$

The adversarial margin measures the gap between the model's confidence in the correct category and the strongest confusable category. The smaller the value, the more probably $x_i^{adv}$ is to be misclassified. A negative margin indicates that the attack has been successful.

Therefore, we calculate the average negative adversarial margin for each category:

$$\mathcal{D}c = \frac{1}{N_c} \sum_{i: y_i = c} (-m_i), \tag{2}$$

where $N_c$ means the number of samples in category $c$. This value is used to measure the robustness vulnerability of the category. A higher value indicates that the category $c$ is more likely to be misclassified when facing adversarial attack.

Finally, we apply softmax function to transform $\mathcal{D}c$ into sampling probability distribution $p_c$, which is used as the sampling weight for categories in the subsequent sample generation stage, enabling to adaptively generate more samples of less robust categories. Furthermore, we compare several other reweighting strategies. The experimental results are shown in Appendix E.

## 4.3 Non-Robust Feature Suppression for Balanced Representations

To achieve the fair tendency to each target when generating adversarial examples, we design a FAEs generation method and use them as benign samples, which ensures that FAEs' non-robust feature

predictions are not concentrated in a few categories; otherwise, adversarial perturbations would be more likely to attack such categories.

We adopt a modified information bottleneck approach to achieve this. The standard information bottleneck seeks to get a compressed representation $Z$ of the input $X$ that is maximally informative about the target label $Y$. The objective function can be written as follows:

$$\mathcal{L}_{IB} = \mathcal{I}(Z; Y) - \beta \mathcal{I}(Z; X), \tag{3}$$

where $\mathcal{I}$ denotes the mutual information and $\beta$ balances the trade-off between prediction accuracy and compression. We attempt to distill non-robust features $Z_{nr}$ from the intermediate feature representation $Z = f_l^T(x)$, where $f_l^T(\cdot)$ describes $l$-th layer outputs of the teacher.

When the synthetic samples $x_i$ is input into $f^T$, we inject a learnable noise scale $\lambda_r$ into $Z$ and define informative features $Z_I$ as follows:

$$Z_I = f_l^T(x_i) + \text{softplus}(\lambda_r) \cdot \epsilon, \epsilon \sim \mathcal{N}(0, I). \tag{4}$$

After obtaining $Z_I$ added by $\lambda_r$, we propagate $Z_I$ to the subsequent output layer $f_{l+}^T$ and evaluate the influence of each unit's feature to the teacher prediction. When distilling $Z_{nr}$ from $Z_I$, we must ensure $Z_I$ remains predictive while encouraging robustness to noise. Since directly optimizing mutual information is intractable, we use a variational approximation (Kim et al., 2021). The optimization objective is formulated as:

$$\min \mathcal{L}(\lambda_r) = CE(f_{l+}^T(Z_I), y_i) + \beta \cdot \sum_{c=1}^{Channel} \left( \frac{v_c}{\lambda_c^2} + \log\left(\frac{\lambda_c^2}{v_c}\right) - 1 \right), \tag{5}$$

where $v_c$ and $\lambda_c$ denote the $c$-th channel of $Z_I$ and $\lambda_r$ respectively. The first term ensures that $Z_I$ correctly predicts the target label $y_i$, while the second term acts as a regularizer to control the amount of information passing through the bottleneck.

After optimizing $\lambda_r$, we identify non-robust channels index $i_{nr_k}$ by comparing $\lambda_r^2$ with the maximum variance across all channels:

$$i_{nr_k} = \mathbb{1}\left[ \lambda_k^2 < \max_{c' \in \{1, \ldots, C\}} \left\{ \text{Var}\left( Z_I^{c'} \right) \right\} \right]. \tag{6}$$

The right-hand side represents the upper limit of perturbation. Correspondingly, non-robust features $Z_{nr}$ are obtained via channel-wise masking: $Z_{nr} = i_{nr} \cdot Z$. The prediction results of $Z_{nr}$ are vulnerable to perturbations, so that they are highly correlated with adversarial predictions. To enable the generator to synthesize FAEs, we minimize the $KL$ divergence between the predictions of non-robust features and the uniform distribution:

$$\mathcal{L}_{uni} = KL(\mathcal{U}, f_{l+}^T(Z_{nr})). \tag{7}$$

To further enhance the quality and diversity of the FAEs, we introduce additional loss functions during the training process of the generator:

$$\begin{cases} \mathcal{L}_{adv} = KL\left( f^T(x_i), f^S(x_i) \right) \\ \mathcal{L}_{bn} = \sum_l \left( \|\mu_l(x_i) - \mu_l\|_2 + \|\sigma_l^2(x_i) - \sigma_l^2\|_2 \right), \\ \mathcal{L}_{oh} = CE\left( f^T(x_i), y_i \right) \end{cases} \tag{8}$$

where $\mathcal{L}_{adv}$ encourages divergence between the student and teacher, promoting the diversity of FAEs; $\mathcal{L}_{bn}$ improves the visualization of the FAEs by matching the statistical information (mean $\mu_l$ and variance $\sigma_l^2$) stored in BatchNorm layers of the teacher and the student; $\mathcal{L}_{oh}$ ensures that the FAEs are correctly predicted by the teacher.

Therefore, the overall loss function for the generator in the generation stage are summarized as:

$$\mathcal{L}_{gen} = \lambda_{adv} \cdot \mathcal{L}_{adv} + \lambda_{bn} \cdot \mathcal{L}_{bn} + \lambda_{oh} \cdot \mathcal{L}_{oh} + \lambda_{uni} \cdot \mathcal{L}_{uni}, \tag{9}$$

where these hyperparameters $\lambda_{adv}$, $\lambda_{bn}$, $\lambda_{oh}$ and $\lambda_{uni}$ are adjusted empirically to balance the trade-offs between robustness, fairness, and accuracy. The hyper-parameter selection experiments are shown in Appendix F. By training with the above loss function, the generator synthesizes FAEs $x_F$, which not only convey effective knowledge, but also be fairer in terms of the tendency towards different categories when generating adversarial examples.

Table 1: Result of average robustness(%) (Avg. ↑), worst robustness(%) (Worst ↑), and normalized standard deviation (NSD ↓) on CIFAR-10. RN-18 and MN-V2 are abbreviations of ResNet-18 and MobileNet-V2 respectively. The best results are **bolded**, and the second best results are underlined.

| Student | Method | Clean | | | FGSM | | | PGD | | | CW$_\infty$ | | | AA | | |
|---|---|---|---|---|---|---|---|---|---|---|---|---|---|---|---|---|
| | | Avg. | Worst | NSD | Avg. | Worst | NSD | Avg. | Worst | NSD | Avg. | Worst | NSD | Avg. | Worst | NSD |
| RN-18 | ZSKT | 65.48 | 32.20 | 0.266 | 29.80 | 5.00 | 0.563 | 22.88 | 3.40 | 0.624 | 18.48 | 2.60 | 0.696 | 17.16 | 2.20 | 0.740 |
| | CMI | 68.29 | 51.20 | 0.161 | 33.34 | 19.50 | 0.337 | 28.94 | 15.80 | 0.393 | 27.63 | 14.90 | 0.409 | 26.85 | 14.60 | 0.419 |
| | DeepInv | 71.22 | 58.90 | 0.124 | 38.60 | 22.50 | 0.303 | 27.95 | 15.00 | 0.375 | 27.23 | 14.50 | 0.379 | 25.19 | 11.60 | 0.422 |
| | Fast | 72.20 | 57.80 | 0.132 | 43.96 | 26.90 | 0.295 | 39.64 | 20.10 | 0.339 | 33.96 | 18.40 | 0.388 | 33.24 | 17.70 | 0.401 |
| | DERD | 73.91 | 53.70 | 0.158 | 39.86 | 22.90 | 0.341 | 30.55 | 13.90 | 0.417 | 29.98 | 13.60 | 0.421 | 28.61 | 11.80 | 0.443 |
| | DFARD | 67.93 | 45.90 | 0.247 | 41.27 | 18.40 | 0.458 | 37.34 | 15.20 | 0.491 | 35.47 | 13.30 | 0.513 | 34.48 | 12.00 | 0.552 |
| | DFHL | 74.42 | 58.60 | 0.129 | 41.67 | 24.30 | 0.298 | 36.99 | 19.30 | 0.344 | 37.33 | 17.80 | 0.348 | 36.39 | 18.50 | 0.351 |
| | FERD | **79.26** | **65.90** | **0.105** | **51.09** | **31.30** | **0.237** | **42.24** | **20.40** | **0.331** | **41.73** | **19.30** | **0.339** | **40.12** | **20.80** | **0.325** |
| MN-V2 | ZSKT | 54.69 | 16.10 | 0.394 | 23.62 | 3.90 | 0.663 | 19.83 | 3.00 | 0.702 | 15.86 | 1.90 | 0.725 | 14.85 | 1.60 | 0.749 |
| | CMI | 60.70 | 38.60 | 0.205 | 27.90 | 12.50 | 0.393 | 19.70 | 8.50 | 0.482 | 19.57 | 8.10 | 0.461 | 18.09 | 7.60 | 0.506 |
| | DeepInv | 62.77 | 46.00 | 0.170 | 31.66 | 14.50 | 0.306 | 19.13 | 7.00 | 0.403 | 19.01 | 6.90 | 0.407 | 15.39 | 5.40 | 0.471 |
| | Fast | 58.60 | 43.40 | 0.198 | 29.91 | 15.70 | 0.374 | 24.28 | 11.60 | 0.444 | 19.46 | 7.50 | 0.535 | 18.35 | 6.80 | 0.563 |
| | DERD | 69.80 | 48.70 | 0.175 | 39.99 | 22.10 | 0.329 | 28.82 | 15.10 | 0.372 | 27.97 | 14.00 | 0.386 | 25.80 | 12.00 | 0.403 |
| | DFARD | 66.32 | 40.10 | 0.273 | 38.71 | 12.60 | 0.519 | 33.84 | 9.20 | 0.562 | 32.76 | 8.70 | 0.575 | 31.81 | 8.10 | 0.584 |
| | DFHL | 70.83 | 50.60 | 0.153 | 37.22 | 16.70 | 0.320 | 33.27 | 14.20 | 0.358 | 33.60 | 14.30 | 0.353 | 32.58 | 13.70 | 0.368 |
| | FERD | **78.34** | **64.10** | **0.108** | **53.04** | **33.40** | **0.229** | **41.63** | **20.80** | **0.304** | **41.43** | **20.50** | **0.314** | **38.06** | **20.30** | **0.349** |

## 4.4 LABEL-SPACE ATTACK DIVERSIFICATION FOR FAIRNESS OPTIMIZATION

To address the problem that the student shows significant differences in defending against adversarial attacks with different targets and is more vulnerable to them from specific categories, we propose a novel adversarial examples generation method, named UTAEs generation.

We aim to construct a more evenly distributed type of adversarial perturbation, so that the adversarial targets are not limited to certain "easily misclassified" classes, but uniformly cover the entire class space. To achieve this, we apply a uniform target class constraint during adversarial generation, avoiding to attacks from a single direction. The generation formula is as follows:

$$x_U^{t+1} = \Pi_{x_U+\mathcal{S}}\Big(x_U^t + \alpha \cdot \text{sign}\Big(\nabla_{x_U^t}\big[KL\left(f^T(x_i), f^T(x_U^t)\right) - \gamma \cdot KL\left(\mathcal{U}, f^T(x_U^t)\right)\big]\Big)\Big), \quad (10)$$

where $\nabla_{x_U^t}$ denotes the gradient of the entropy loss function w.r.t. the UTAE in step $t$ and $\alpha$ is the step size. By introducing the $KL$ divergence between $f^T(x_U)$ and $\mathcal{U}$ in the adversarial example generation, we make the target distribution of adversarial examples more extensive.

After synthesizing FAEs $x_F$ and UTAEs $x_U$, we employ them as benign samples and adversarial examples respectively for robust distillation. Here, we adopt a decoupled knowledge distillation loss (Cui et al., 2024) instead of the traditional KL divergence:

$$\begin{aligned}
\mathcal{L}_{IKL}(f^T(x_t), f^S(x_s)) = &\frac{1}{4}\left\|\sqrt{\mathcal{S}(\overline{w}_y)}\Big(\Delta f^T(x_t) - \Delta f^S(x_s)\Big)\right\|^2 \\
&- \mathcal{S}\Big(softmax(f^T(x_t))^\top\Big) \cdot \log softmax(f^S(x_s)),
\end{aligned} \quad (11)$$

where $\mathcal{S}(\cdot)$ denotes stop gradients operation, $\overline{w}_y$ represents the class-wise global information weight corresponding to the ground-truth label $y$. $\Delta f^T(x_t)$ and $\Delta f^S(x_s)$ represent pairwise difference matrices, $\Delta f^T(x_t)_{j,k} = f^T(x_t)_j - f^T(x_t)_k$, $\Delta f^S(x_s)_{j,k} = f^S(x_s)_j - f^S(x_s)_k$. The loss function is a composition of a wMSE loss and a Cross-Entropy loss. The first term minimizes the structural difference between the teacher's and student's logits, and the second term aligns the student's predictive distribution with that of the teacher. The decoupled formulation breaks the asymmetric optimization property and performs better in knowledge distillation and adversarial training.

Therefore, the robust distillation framework is as follows:

$$\mathcal{L}_{stu} = \lambda_1 \mathcal{L}_{IKL}\left(f^T(x_F), f^S(x_F)\right) + \lambda_2 \mathcal{L}_{IKL}\left(f^T(x_F), f^S(x_U)\right). \quad (12)$$

Through robust distillation of diversified adversarial targets, we force the student to inherit robustness across a far broader range of categories, thereby enhancing overall defensive capability against attacks from all directions.

Table 2: Result of average robustness(%) (Avg. ↑), worst-10% robustness(%) (Worst ↑), and normalized standard deviation (NSD ↓) on CIFAR-100. RN-18 and MN-V2 are abbreviations of ResNet-18 and MobileNet-V2. The best results are **bolded**, and the second best results are underlined.

| Student | Method | Clean | | | FGSM | | | PGD | | | CW$_\infty$ | | | AA | | |
|---|---|---|---|---|---|---|---|---|---|---|---|---|---|---|---|---|
| | | Avg. | Worst | NSD | Avg. | Worst | NSD | Avg. | Worst | NSD | Avg. | Worst | NSD | Avg. | Worst | NSD |
| RN-18 | ZSKT | 36.44 | 6.40 | 0.522 | 12.69 | 0.10 | 0.985 | 10.17 | 0.00 | 1.083 | 7.12 | 0.00 | 1.273 | 6.01 | 0.00 | 1.361 |
| | CMI | 51.60 | 27.20 | 0.295 | 25.40 | 5.60 | 0.624 | 22.63 | 3.50 | 0.686 | 18.94 | 1.90 | 0.785 | 17.72 | 1.30 | 0.824 |
| | DeepInv | 53.44 | 27.20 | 0.298 | 26.14 | 5.60 | 0.630 | 23.11 | 3.60 | 0.699 | 19.05 | 2.20 | 0.792 | 17.49 | 1.60 | 0.857 |
| | Fast | 50.69 | 23.50 | 0.323 | 26.45 | 5.50 | 0.634 | 24.13 | 4.50 | 0.687 | 19.44 | 2.20 | 0.806 | 18.24 | **1.70** | 0.859 |
| | DERD | 54.01 | 28.10 | 0.295 | 27.83 | 5.20 | 0.631 | 24.78 | 4.70 | 0.689 | 22.56 | 3.20 | 0.776 | 21.01 | 1.40 | 0.856 |
| | DFARD | 50.43 | 21.80 | 0.340 | 26.44 | 4.80 | 0.644 | 24.22 | 3.90 | 0.693 | 21.56 | 2.50 | 0.781 | 20.21 | 1.50 | 0.828 |
| | DFHL | 46.27 | 18.70 | 0.370 | 18.73 | 2.60 | 0.799 | 15.72 | 1.30 | 0.933 | 16.33 | 1.50 | 0.896 | 15.23 | 1.10 | 0.954 |
| | FERD | **55.91** | **29.00** | **0.289** | **29.23** | **5.70** | **0.622** | **26.10** | **4.90** | **0.681** | **23.39** | **3.30** | **0.754** | **21.79** | **1.70** | **0.805** |
| MN-V2 | ZSKT | 30.79 | 2.70 | 0.650 | 13.66 | 0.00 | 0.975 | 12.44 | 0.00 | 1.007 | 9.54 | 0.00 | 1.124 | 8.62 | 0.00 | 1.208 |
| | CMI | 35.80 | 6.00 | 0.425 | 13.82 | 0.60 | 0.828 | 9.91 | 0.40 | 0.966 | 9.06 | 0.00 | 1.000 | 8.04 | 0.00 | 1.042 |
| | DeepInv | 38.10 | 8.00 | 0.447 | 13.97 | 1.00 | 0.920 | 9.98 | 1.00 | 1.123 | 8.12 | 0.00 | 1.260 | 6.67 | 0.00 | 1.416 |
| | Fast | 39.00 | 6.00 | 0.441 | 16.42 | 1.40 | 0.821 | 14.80 | 0.80 | 0.913 | 10.44 | 0.00 | 1.085 | 9.53 | 0.00 | 1.139 |
| | DERD | 54.12 | 29.10 | 0.291 | 26.31 | 5.40 | 0.646 | 22.39 | 3.00 | 0.742 | 20.59 | 2.70 | 0.774 | 18.92 | 1.70 | 0.832 |
| | DFARD | 49.02 | 20.60 | 0.340 | 24.62 | 4.00 | 0.668 | 22.42 | **3.10** | 0.722 | 20.65 | 2.10 | 0.778 | 19.30 | 1.60 | 0.826 |
| | DFHL | 42.08 | 11.30 | 0.436 | 16.25 | 0.90 | 0.914 | 13.55 | 0.40 | 1.015 | 14.06 | 0.70 | 0.980 | 12.99 | 0.40 | 1.051 |
| | FERD | **56.14** | **29.20** | **0.285** | **29.63** | **6.10** | **0.611** | **24.48** | **3.10** | **0.713** | **23.20** | **3.30** | **0.743** | **21.20** | **2.30** | **0.793** |

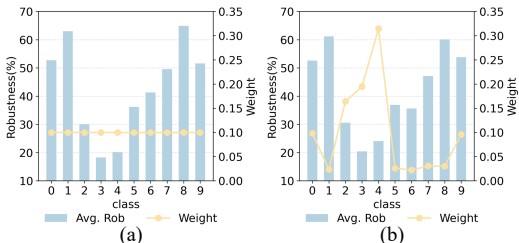

Figure 4: (a): The robustness of the student for each category under equal weight. (b): The robustness of the student for each category under reweighting situation.

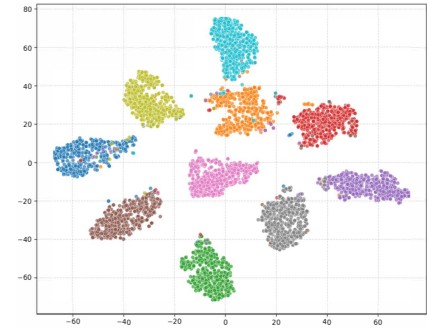

Figure 5: t-SNE plot for illustrating the FAEs representations in feature space.

## 5 EXPERIMENTS

### 5.1 EXPERIMENTAL SETTINGS

**Datasets & Models.** We conduct our experiments on three datasets: CIFAR-10 (Krizhevsky et al., 2009), CIFAR-100, and Tiny-ImageNet (Le & Yang, 2015). For the teacher model, we select WideResNet-34-10 (Zagoruyko & Komodakis, 2016) for both CIFAR-10 and CIFAR-100, while PreActResNet-34 is used for Tiny-ImageNet. For the student model, we select ResNet-18 (He et al., 2016a) and MobileNet-v2 (Sandler et al., 2018) for CIFAR-10 and CIFAR-100. For Tiny-ImageNet, we use PreActResNet-18. The performance of the teacher and the experiments on Tiny-ImageNet are shown in Appendix C and D.

**Baselines.** We compare our method with seven different DFRD methods, including ZSKT (Micaelli & Storkey, 2019), CMI (Fang et al., 2021), DeepInv (Yin et al., 2020), Fast (Fang et al., 2022), DERD (Zhou et al., 2024), DFARD (Wang et al., 2024a) and DFHL (Yuan et al., 2024). Note that the first four methods belong to DFKD initially and do not involve robustness. We generate adversarial examples and then apply the same distillation training loss as RSLAD (Zi et al., 2021) to transform them into DFRD.

**Implementation Details.** Our proposed method and all baselines are implemented on NVIDIA A800 GPU. The generator is trained via Adam optimizer with a learning rate of 2e-3, $\beta_1$ of 0.5, $\beta_2$ of 0.999. The student is trained via SGD optimizer with an initial value of 0.1, momentum of 0.9, and weight decay of 5e-4. The distillation epochs is set to 220 and the training iterations of generator and student are 200 and 400 respectively. The batch size is set to 256 for CIFAR-10 and 512 for both CIFAR-100 and Tiny-ImageNet. In Eq.4, we extract the intermediate features from the

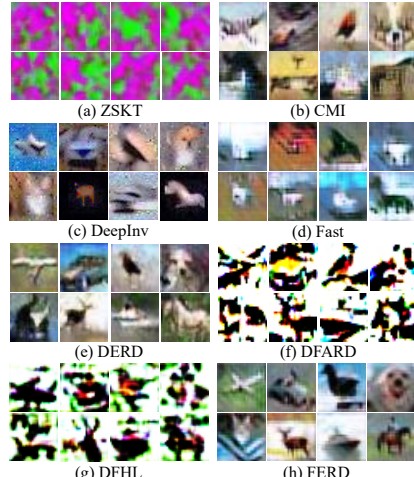

Figure 6: 32×32 images generated by inverting a WideResNet-34-10 trained on CIFAR-10.

Table 3: Result of average robustness(%) (Avg. ↑), worst robustness(%) (Worst ↑) on CIFAR-10. The teacher model is WideResNet-34-20 and the student model is ResNet-18.

| Method | Clean | | AA | |
|---|---|---|---|---|
| | Avg. | Worst | Avg. | Worst |
| ZSKT | 50.14 | 14.20 | 3.39 | 0.20 |
| CMI | 73.26 | 57.90 | 26.44 | 9.80 |
| DeepInv | 69.47 | 56.20 | 24.80 | 13.40 |
| Fast | 68.95 | 53.30 | 30.28 | 14.70 |
| DERD | 69.00 | 45.70 | 33.50 | 15.10 |
| DFARD | 70.81 | 49.10 | 34.38 | 14.60 |
| DFHL | 70.89 | 52.60 | 36.59 | 15.60 |
| FERD | **78.23** | **61.70** | **38.67** | **16.00** |

last convolutional layer $l$. In Eq.9, Eq.10, and Eq.12, we set the hyperparameter $\lambda_{adv}$=1, $\lambda_{bn}$=5, $\lambda_{oh}$=1, $\lambda_{uni}$=5, $\gamma$=0.5, $\lambda_1$=5/6, and $\lambda_2$=1/6.

**Evaluation Metrics.** We evaluate the robustness of the student against four adversarial attacks: FGSM, PGD, CW$_\infty$, and AA. Following (Yue et al., 2023; Zhao et al., 2024), we employ the worst-class robustness and Normalized Standard Deviation (NSD) to quantify the robust fairness across categories. NSD is a normalized metric of the adversarial robustness with respect to the standard deviation across different classes. The smaller the value of NSD, the better. Notably, for CIFAR-100 and Tiny-ImageNet, we adopt worst-10% robustness in place of worst-class robustness, due to the limited size of test set per category and poor performance in the worst class robustness.

## 5.2 EXPERIMENTAL RESULTS

**Overall performance.** Tab. 1 and Tab. 2 show the performance of ResNet-18 and MobileNet-V2 distilled on CIFAR-10 and CIFAR-100 by our method and baselines. The results demonstrate that the student distilled from FERD has an improvement in the robustness and fairness. Our method achieves state-of-the-art worst-class robustness under all attacks. When distilled on CIFAR-10 with MobileNet-V2, FERD improves the worst class robustness by 11.3%, 5.7%, 6.2%, and 6.6% compared with the best baseline against four adversarial attack respectively. In addition, there is an improvement in the average accuracy and NSD in most situations. Specifically, as shown in Tab. 1, FERD achieves the highest average accuracy under all attacks with an improvement of up to 13.05%, which is attributed to our high-quality synthetic samples.

**Effectiveness of reweighting.** Fig. 4 illustrates the impact of reweighting sampling strategy on the robustness of the student model. The robustness-guided reweighting strategy we proposed effectively identifies the categories with poor robustness and increases their sampling weights. Specifically, the weight of the fourth class which has the lowest robustness reaches 0.314, exceeding other classes substantially. Correspondingly, the student's robustness in these categories are improved, thereby enhancing robust fairness.

**Synthetic data visualization.** In Fig. 6, we visualize the synthetic data generated by FERD and baselines. The results indicate that our generator is able to synthesize more visible data. In such scenarios, CMI and Fast even suffer from model collapse problem, where the visual quality of the synthetic samples is extremely low. This further demonstrates the superiority of our method, which recovers high-quality samples from the robust teacher. In Fig. 5, we find that the points of different colors are clearly distributed in different areas, indicating that the class distinction is excellent.

**Distillation with different teacher.** To verify the transferability of our method, we select the WRN-34-20 model as the teacher model for experiments. As shown in the Tab. 3, FERD still outperforms

Table 4: Ablation study of different components in the framework.

| Method | Clean | | AA | |
|---|---|---|---|---|
| | Avg. | Worst | Avg. | Worst |
| FERD | 79.26 | **65.90** | 42.24 | **20.80** |
| w/o reweighting | **79.53** | 63.20 | **42.54** | 17.20 |
| w/o FAEs | 78.42 | 65.30 | 41.01 | 19.90 |
| w/o UTAEs | 77.57 | 65.10 | 41.24 | 20.00 |
| w/o FAEs UTAEs | 77.48 | 62.50 | 40.92 | 20.30 |

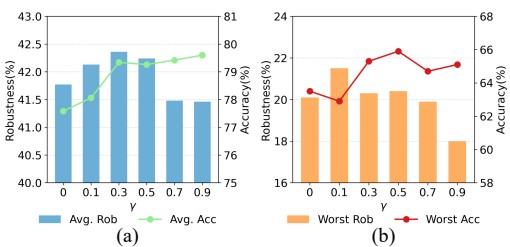

Figure 7: Ablation study. (a): The average robustness and accuracy of the student under different $\gamma$. (b):The worst-class robustness and accuracy of the student under different $\gamma$.

all baselines when distilled from the WRN-34-20 teacher. Specifically, FERD achieves the highest average robustness and worst-robustness, achieving the improvement of 2.08% and 0.4% respectively under AutoAttack. This is attributed to our designed loss functions in Section 4.3, which can invert high-quality images from teacher models with different architectures.

## 5.3 ABLATION STUDY

In this section, we provide ablation studies on FERD. We keep the same settings with experiments and use WideResNet-34-10 as the teacher, ResNet-18 as the student and CIFAR-10 as dataset.

**Effectiveness of all components.** To verify the effectiveness of our FERD, we conduct ablation studies for each component, and the experimental results are shown in Tab. 4. We first examine the impact of the reweighting strategy. Compared with the absence of it, the student's accuracy on worst-class significantly improves when reweighting strategy is applied. However, it slightly compromises the overall accuracy, which is consistent with the findings of previous researches (Yue et al., 2023; Zhao et al., 2024). Further, we analyze the contributions of FAEs and UTAEs. Removing either component individually results in a consistent performance decline across all metrics. When both FAEs and UTAEs are removed simultaneously, the performance further deteriorates. This confirms that FAEs and UTAEs are complementary and their joint effect is crucial for achieving the best robust fairness performance.

**Hyperparameter $\gamma$.** In Fig.7 (a) and (b), we illustrate the average and worst-class performance of the student distilled with varying $\gamma$ during UTAEs generation. $\gamma$ controls the strength of the uniform target class constraint during adversarial examples generation. Note that when $\gamma=0$, the adversarial examples are the same as the standard PGD method. For average robustness and average accuracy, we observe that a low to medium $\gamma$ (e.g., 0.1 to 0.5) positively enhances the robustness and fairness of the student. This indicates the uniform constraint enhances the robustness of the model without significantly affecting the intensity of adversarial perturbations. However, A high $\gamma$ (e.g., 0.7 or 0.9) enforces strong uniform constraint, which overly suppress the optimization of the adversarial loss, leading to weaker perturbations that reduce attack strength of the adversarial examples. This negatively affects the robust distillation and leads to a reduction in overall robustness.

## 6 CONCLUSION

In this paper, we made the first attempt to investigate the robust fairness in DFRD. We summarized two key factors affecting robust fairness and propose a FERD framework to mitigate these problems by adjusting the proportion and distribution of adversarial examples. For the proportion, we introduced a robustness-guided class reweighting strategy to encourage the generator to synthesize more samples from weakly robust classes. For the distribution, we designed FAEs and UTAEs, taking them as benign samples and adversarial examples respectively for robust distillation. Extensive experiments show that FERD significantly improves the robust and fairness performance of the student model. Our work is more applicable and can be effectively applied in practical scenarios.

## ACKNOWLEDGMENT

Zhengxiao Li, Liming Lu, and Shuchao Pang are supported by National Key Research and Development Program of China under (Grant No.2023YFB2703900), the National Natural Science Foundation of China (Grant No.62206128) and the Postgraduate Research & Practice Innovation Program of Jiangsu Province (Grant No.KYCX24_0723). Yongbin Zhou is supported by the National Natural Science Foundation of China (Grant No.U2336205).

## ETHICS STATEMENT

This paper aims to address the issue of robust fairness in data-free robustness distillation. The FERD framework we propose is designed to enhance the fairness of the model's robust performance across different categories, thereby improving its reliability and security in real-world applications. All the experiments are conducted on publicly available and standard datasets, which are widely used in the machine learning. These datasets do not involve any sensitive or private personal information, avoiding the risks of data privacy and abuse.

## REPRODUCIBILITY STATEMENT

To ensure the reproducibility of our work, we have provided comprehensive details of our experiments. The experimental setting of FERD is detailed in Section 5.1, including the models, baselines, evaluation metrics, and implementation details. All datasets (CIFAR-10, CIFAR-100, and Tiny-ImageNet) are publicly available. Final hyperparameter values for all experiments are explicitly listed, with further analysis on their selection provided in Appendices F. To facilitate direct replication of our results, we make our source code publicly available upon publication.

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

## A PROOFS

### A.1 THE PROOF OF CONJECTURE 1

Conjecture 1 is: by adjusting the weight of class sampling, the problem of robust unfairness is effectively alleviated.

**Assumption 1.** The robust risk function $\mathcal{R}_c(\theta)$ for each class $c$ is differentiable and $L$-smooth (Bottou et al., 2018). That is, for any $\theta_1, \theta_2$, there exists a constant $L > 0$ such that:

$$\mathcal{R}_c(\theta_2) \leq \mathcal{R}_c(\theta_1) + \langle \nabla \mathcal{R}_c(\theta_1), \theta_2 - \theta_1 \rangle + \frac{L}{2} \|\theta_2 - \theta_1\|^2. \tag{13}$$

**Theorem 1.** *If the sampling weight $p_k$ for the weak-robust class $k$ satisfies $p_k > \frac{1}{C}$ at step $t$, then the expected risk reduction for the weak-robust class $k$ is strictly greater under reweighting sampling than under uniform sampling, provided that the gradient alignment condition holds:*

$$\mathbb{E}[\mathcal{R}_k(\theta_{t+1}^{rew})] < \mathbb{E}[\mathcal{R}_k(\theta_{t+1}^{uni})]. \tag{14}$$

*Proof.* Consider the SGD update rule at step $t$. For uniform sampling, the expected update direction is $\mathbb{E}[\Delta \theta_{uni}] = -\eta \frac{1}{C} \sum_c \nabla \mathcal{R}_c(\theta_t)$. For reweighting sampling, the expected update direction is $\mathbb{E}[\Delta \theta_{rew}] = -\eta \sum_c p_c \nabla \mathcal{R}_c(\theta_t)$.

Using the $L$-smoothness on the weak-robust class $k$:

$$\mathcal{R}_k(\theta_{t+1}) \leq \mathcal{R}_k(\theta_t) + \langle \nabla \mathcal{R}_k(\theta_t), \Delta \theta \rangle + \frac{L}{2} \|\Delta \theta\|^2. \tag{15}$$

We compare the linear decrement term $\mathcal{D} = \langle \nabla \mathcal{R}_k(\theta_t), \Delta \theta \rangle$ for both strategies, neglecting the second-order term for sufficiently small $\eta$.

For uniform sampling:

$$\mathcal{D}_{uni} = -\eta \langle \nabla \mathcal{R}_k, \frac{1}{C} \sum_{c=1}^{C} \nabla \mathcal{R}_c \rangle = -\eta \left( \frac{1}{C} \|\nabla \mathcal{R}_k\|^2 + \frac{1}{C} \sum_{c \neq k} \langle \nabla \mathcal{R}_k, \nabla \mathcal{R}_c \rangle \right). \tag{16}$$

For reweighting sampling:

$$\mathcal{D}_{rew} = -\eta \langle \nabla \mathcal{R}_k, \sum_{c=1}^{C} p_c \nabla \mathcal{R}_c \rangle = -\eta \left( p_k \|\nabla \mathcal{R}_k\|^2 + \sum_{c \neq k} p_c \langle \nabla \mathcal{R}_k, \nabla \mathcal{R}_c \rangle \right). \tag{17}$$

We aim to prove $\mathcal{D}_{rew} < \mathcal{D}_{uni}$. Let $\phi_{k,c} = \langle \nabla \mathcal{R}_k, \nabla \mathcal{R}_c \rangle$ and we analyze the difference:

$$\Delta_{diff} = \mathcal{D}_{uni} - \mathcal{D}_{rew} = \eta \left( (p_k - \frac{1}{C}) \|\nabla \mathcal{R}_k\|^2 + \sum_{c \neq k} (p_c - \frac{1}{C}) \phi_{k,c} \right). \tag{18}$$

Reweighting strategy assigns $p_k \gg \frac{1}{C}$ for the weak-robust class $k$, and $p_c < \frac{1}{C}$ for robust classes. Since $\|\nabla \mathcal{R}_k\|^2 > 0$ is the dominant gradient term for the weak-robust class, the term $(p_k - \frac{1}{C}) \|\nabla \mathcal{R}_k\|^2$ is positive and large.

While gradient conflicts may exist ($\phi_{k,c} < 0$), the gradients of vulnerable classes typically exhibit larger magnitudes compared to robust classes due to their higher loss values in adversarial training(Madry et al., 2017). Specifically, the term $(p_k - \frac{1}{C}) \|\nabla \mathcal{R}_k\|^2$ outweighs the residual terms. Thus, $\Delta_{diff} > 0$, implying $\mathcal{D}_{fer} < \mathcal{D}_{uni}$.

Consequently, the upper bound of the worst-class risk decreases more significantly under reweighting sampling:

$$\mathcal{R}_k(\theta_{t+1}^{rew}) \leq \mathcal{R}_k(\theta_t) + \mathcal{D}_{rew} + O(\eta^2) < \mathcal{R}_k(\theta_t) + \mathcal{D}_{uni} + O(\eta^2). \tag{19}$$

Then the Theorem 1 is proved.

## A.2 THE PROOF OF CONJECTURE 2

Conjecture 2 is: robust distillation on adversarial examples with uniformly distributed targets defend against attacks from different targets.

**Definition 1.** The standard adversarial risk makes the worst perturbation within a ball $\mathcal{B}_\epsilon(x) = \{x' : \|x' - x\|_p \leq \epsilon\}$:

$$\mathcal{R}_{std}(\theta) = \mathbb{E}_{(x,y)\sim\mathcal{D}} \left[ \max_{\delta \in \mathcal{B}_\epsilon(0)} \mathcal{L}(f_\theta(x + \delta), y) \right]. \tag{20}$$

Adversarial attacks implicitly targets the decision boundary of the nearest incorrect class.

**Definition 2.** Adversarial examples with uniformly distributed targets enforce attacks towards a target class $t$ sampled uniformly from $\mathcal{Y} \setminus \{y\}$. The risk objective is:

$$\mathcal{R}_{uni}(\theta) = \mathbb{E}_{(x,y)\sim\mathcal{D}} \left[ \mathbb{E}_{t\sim\mathcal{U}(\mathcal{Y}\setminus\{y\})} \left[ \max_{\delta \in \mathcal{B}_\epsilon(0)} \mathcal{L}_{target}(f_\theta(x + \delta), t) \right] \right], \tag{21}$$

where $\mathcal{L}_{target}$ encourages the model to classify $x + \delta$ as $t$.

**Assumption 2.** The event of a successful untargeted adversarial attack is the union of successful targeted attacks towards all other classes $k \neq y$. Specifically, let $E$ be the error region $E = \{x' \in \mathcal{B}_\epsilon(x) : \arg\max f_\theta(x') \neq y\}$. We assume $E = \bigcup_{k\neq y} E_k$, where $E_k$ is the region where the input is misclassified specifically as class $k$ (Carlini & Wagner, 2017).

**Theorem 2.** *Minimizing the Uniform-Target Risk $\mathcal{R}_{uni}(\theta)$ minimizes an upper bound of the Standard Adversarial Risk $\mathcal{R}_{std}(\theta)$.*

*Proof.* The standard adversarial loss seeks the perturbation that maximizes the loss, corresponding to the model predicting an incorrect class $k$:

$$\max_\delta \mathcal{L}(f_\theta(x + \delta), y) \approx \max_{k\neq y} \left( \max_\delta \mathcal{L}_{target}(f_\theta(x + \delta), k) \right). \tag{22}$$

Let $\ell_k(\theta, x) = \max_{\delta \in \mathcal{B}_\epsilon} \mathcal{L}_{target}(f_\theta(x + \delta), k)$ be the worst loss towards specific target $k$. The standard robust risk is determined by the maximum component:

$$\mathcal{L}_{std}(\theta, x) = \max_{k\neq y} \ell_k(\theta, x). \tag{23}$$

The training on adversarial examples with uniformly distributed targets minimizes the expectation as follows:

$$\mathcal{L}_{uni}(\theta, x) = \frac{1}{C - 1} \sum_{k\neq y} \ell_k(\theta, x). \tag{24}$$

Therefore, we have the following inequality:

$$\max_{k\neq y} \ell_k(\theta, x) \leq \sum_{k\neq y} \ell_k(\theta, x) = (C - 1) \cdot \mathcal{L}_{uni}(\theta, x). \tag{25}$$

Taking the expectation over the data distribution $\mathcal{D}$:

$$\mathcal{R}_{std}(\theta) \leq (C - 1) \cdot \mathcal{R}_{uni}(\theta). \tag{26}$$

This inequality proves that training on adversarial examples with uniformly distributed targets minimizes a upper bound of the standard adversarial risk. Then the Theorem 2 is proved.

## B THE WHOLE FRAMEWORK OF FERD

This section provides the whole framework of our Fairness-Enhanced data-free adversarial Robust Distillation (FERD). The FERD framework is composed of three key components: a robustness-guided class reweighting strategy, the generation of Fairness-Aware Examples (FAEs), and the construction of Uniform-Target Adversarial Examples (UTAEs). The detailed algorithm description is outlined in Algorithm 1.

---

**Algorithm 1:** The Whole Framework of FERD

---

**Input:** A pre-trained robust teacher $f^T$, a student $f^S$ with parameter $\theta_s$, a generator $g$ with parameter $\theta_g$, distillation epochs $T$, the iterations of generator $g$ in each epoch $T_g$, the iterations of student $f^S$ in each epoch $T_s$, class sampling probability distribution $p$, learning rate for generator $\eta_g$ and student $\eta_s$.

1   Randomly initialize parameters $\theta_g$ and $\theta_s$ ;
2   Uniformly initialize $p$ ;
3   Initialize an empty dataset $D = \{\}$ ;
4   **for** *number of iterations $T$* **do**
5      //∗ Generation stage ∗//
6      $z \sim \mathcal{N}(0,1), y \sim p$ ;
7      **for** *number of iterations $T_g$* **do**
8         $x_F \leftarrow g(z,y)$ ;                          `// Synthesize FAEs`
9         $\theta_g \leftarrow \theta_g - \eta_g \cdot \nabla_{\theta_g} \mathcal{L}_{gen}(f^T, f^S, x_F, y)$ ;
10        $x_F^* \leftarrow select(x_F)$ ;
11      $D \leftarrow D \cup \{x_F^*\}$ ;
12      $p \leftarrow reweight(\mathcal{L}_{class})$ ;                      `// Reweighting`
13      //∗ Robustness distillation stage ∗//
14      **for** *number of iterations $T_s$* **do**
15         $x_F \leftarrow sample(D)$ ;
16         $x_U \leftarrow UTAEs(x_F)$ ;                    `// Create UTAEs`
17         $\theta_s \leftarrow \theta_s - \eta_s \cdot \nabla_{\theta_s} \mathcal{L}_{stu}(f^T, f^S, x_F, x_U)$ ;
**Output:** A student $f^S$ with fair robustness

---

## C   THE PERFORMANCE OF THE TEACHER

In this section, we show the performance of the teacher models under various attacks. We select WideResNet-34-10 (Zagoruyko & Komodakis, 2016) trained by (Chen & Lee, 2024) on CIFAR-10 and CIFAR-100 (Krizhevsky et al., 2009), respectively. Meanwhile, we select PreActResNet-34 (He et al., 2016b) trained by TRADES (Zhang et al., 2019) on Tiny-ImageNet (Le & Yang, 2015). The performance is shown in Tab. 5.

Table 5: Robustness (%) of the teacher models.

| Dataset | Model | Clean | FGSM | PGD | CW$_\infty$ | AA |
|---|---|---|---|---|---|---|
| CIFAR-10 | WRN-34-10 | 86.55 | 64.96 | 61.53 | 58.07 | 57.30 |
| CIFAR-100 | WRN-34-10 | 64.32 | 39.51 | 36.72 | 32.56 | 31.15 |
| Tiny-ImageNet | PARN-34 | 48.96 | 25.69 | 23.47 | 19.56 | 18.56 |

## D   THE RESULTS ON TINY-IMAGENET

We conduct experiments on Tiny-ImageNet. We select PreActResNet-34 as the teacher and PreActResNet-18 as the student. For the settings, we keep consistent with the experiments on CIFAR-100. The results are shown in Tab. 6. The results prove the effectiveness of FERD, achieving state-of-the-art accuracy and worst-10% robustness on Tiny-ImageNet. Note that ZSKT is not applicable for large datasets and the distilled student performs extremely poorly on Tiny-ImageNet, so we do not compare with it on this dataset. The worst-10% robustness under both CW$_\infty$ and AA attacks is all 0, so we also do not compare on it.

From the results, it is observed that our proposed FERD achieves optimal performance under most adversarial attacks. Specifically, FERD achieves state-of-the-art average robustness and NSD under all attacks. For the average robustness, FERD is 0.34%, 1.50%, 0.39%, and 0.40% higher than the suboptimal method under the four adversarial attacks. For the NSD, FERD also comprehensively improves 0.015, 0.004, 0.030 and 0.037 respectively.

Table 6: Result of average robustness(%) (Avg. ↑), worst-10(%) (Worst ↑), and normalized standard deviation (NSD ↓) on Tiny-ImageNet. The best results are **bolded** and the second best results are underlined respectively.

| Method | Clean | | | FGSM | | | PGD | | | CW$_\infty$ | | AA | |
|---|---|---|---|---|---|---|---|---|---|---|---|---|---|
| | Avg. | Worst | NSD | Avg. | Worst | NSD | Avg. | Worst | NSD | Avg. | NSD | Avg. | NSD |
| CMI | 38.56 | 11.90 | 0.445 | 16.46 | 1.10 | 0.785 | 13.43 | 0.00 | 0.882 | 9.93 | 1.109 | 9.23 | 1.152 |
| DeepInv | 36.25 | 9.70 | 0.465 | 13.46 | 0.50 | 0.861 | 10.95 | 0.00 | 0.958 | 7.32 | 1.202 | 6.58 | 1.288 |
| Fast | 33.76 | 7.80 | 0.473 | 12.94 | 0.40 | 0.804 | 11.19 | 0.00 | 0.846 | 7.66 | 1.087 | 7.11 | 1.138 |
| DERD | 36.32 | 9.40 | 0.470 | 16.10 | 1.60 | 0.737 | 14.32 | 0.20 | 0.779 | 10.68 | 0.977 | 9.67 | 1.044 |
| DFARD | 39.37 | 11.60 | 0.448 | 17.66 | 1.30 | 0.750 | 14.44 | 0.30 | 0.804 | 11.80 | 0.967 | 10.84 | 1.046 |
| DFHL | 23.13 | 1.10 | 0.719 | 6.87 | 0.00 | 1.368 | 5.51 | 0.00 | 1.555 | 5.43 | 1.569 | 4.95 | 1.654 |
| FERD | **39.64** | **12.30** | **0.439** | **18.00** | **2.10** | **0.722** | **15.94** | **0.40** | **0.783** | **12.19** | **0.937** | **11.24** | **1.007** |

Table 7: The performance of different reweighting strategies. The best results are **bolded**.

| Method | Clean | | | FGSM | | | PGD | | |
|---|---|---|---|---|---|---|---|---|---|
| | Avg. | Worst | NSD | Avg. | Worst | NSD | Avg. | Worst | NSD |
| power | **80.54** | 65.20 | 0.111 | 52.04 | 28.70 | 0.280 | **42.79** | 17.90 | 0.334 |
| sig | 80.16 | 64.10 | 0.111 | 51.29 | 27.70 | 0.290 | 42.58 | 18.10 | 0.350 |
| linear | 79.89 | 63.70 | 0.113 | **52.06** | 29.10 | 0.275 | 42.23 | 17.10 | 0.344 |
| tanh | 80.40 | 64.60 | 0.115 | 51.47 | 28.90 | 0.284 | 42.30 | 18.20 | 0.345 |
| FERD | 79.26 | **65.90** | **0.105** | 51.09 | **31.30** | **0.237** | 42.24 | **20.40** | **0.331** |

# E COMPARISON OF REWEIGHTING STRATEGIES

To further demonstrate the superiority of our reweighting strategy, we conduct experimental comparisons with different reweighting functions (Yue et al., 2023). We use WideResNet-34-10 as the teacher, ResNet-18 as the student and CIFAR-10 as dataset, while other settings keep consistent with main paper.

The reweighting strategies we compare are all based on a important metric: the least PGD steps (LPS), which represents the steps needed for perturbations of a benign example to become a adversarial example. The average LPS within a class $\kappa_c$ serve as a metric of class-wise robustness , which is calculated as follows:

$$\kappa_c = \frac{1}{N_c} \sum_{j=1}^{N_c} \arg \min_{t \in [0,K]} \left( f^T \left( x_{adv}^{(t)} \right) \neq y \right),$$
(27)

where $K$ is the PGD steps. The higher the value, the more robust the model is to this class, in which case they reduce the weight during training.

Based on the metric, we compare four reweighting strategies. The first is power-type function:

$$\omega_c = \frac{1}{\kappa_c^\beta},$$
(28)

where $\beta$ controls the smoothness of the reweighting function. The second strategy is linear-type function:

$$\omega_c = 1 - \frac{\kappa_c}{K+1}.$$
(29)

The third strategy is sigmoid-type function:

$$\omega_c = sigmoid \left( \lambda + 5 \times \left( 1 - 2 \times \frac{\kappa_c}{K} \right) \right).$$
(30)

The last strategy is tanh-type function:

$$\omega_c = \frac{1 + \tanh \left( \lambda + 5 \times \left( 1 - 2 \times \frac{\kappa_c}{K} \right) \right)}{2},$$
(31)

where $\lambda$ is the parameter. For the above parameters, we set $K = 20$, $\beta = 2$ and $\lambda = 0$ respectively following (Yue et al., 2023). The experimental results are shown in Tab. 7.

We observe that the our reweighting strategy achieves the best results in the aspect of worst-class robustness, which demonstrates that it identifies the categories with poor robustness more effectively.

## F  HYPER-PARAMETER SELECTION

In this section, we show the effect of parameters on the distillation results and illustrate how we determine the choice of parameters. We select WideResNet-34-10 as the teacher and ResNet-18 as the student to experiment on CIFAR-10.

$\lambda_{adv}$, $\lambda_{bn}$, $\lambda_{oh}$ and $\lambda_{uni}$ represent the weights of different loss terms in the loss function $\mathcal{L}_{gen}$, respectively. In order to explore the influence of different weights on the experimental results, we change the weights, respectively, for the experiments. The results are shown in Tab. 8. The corresponding synthetic samples are shown in Fig. 8.

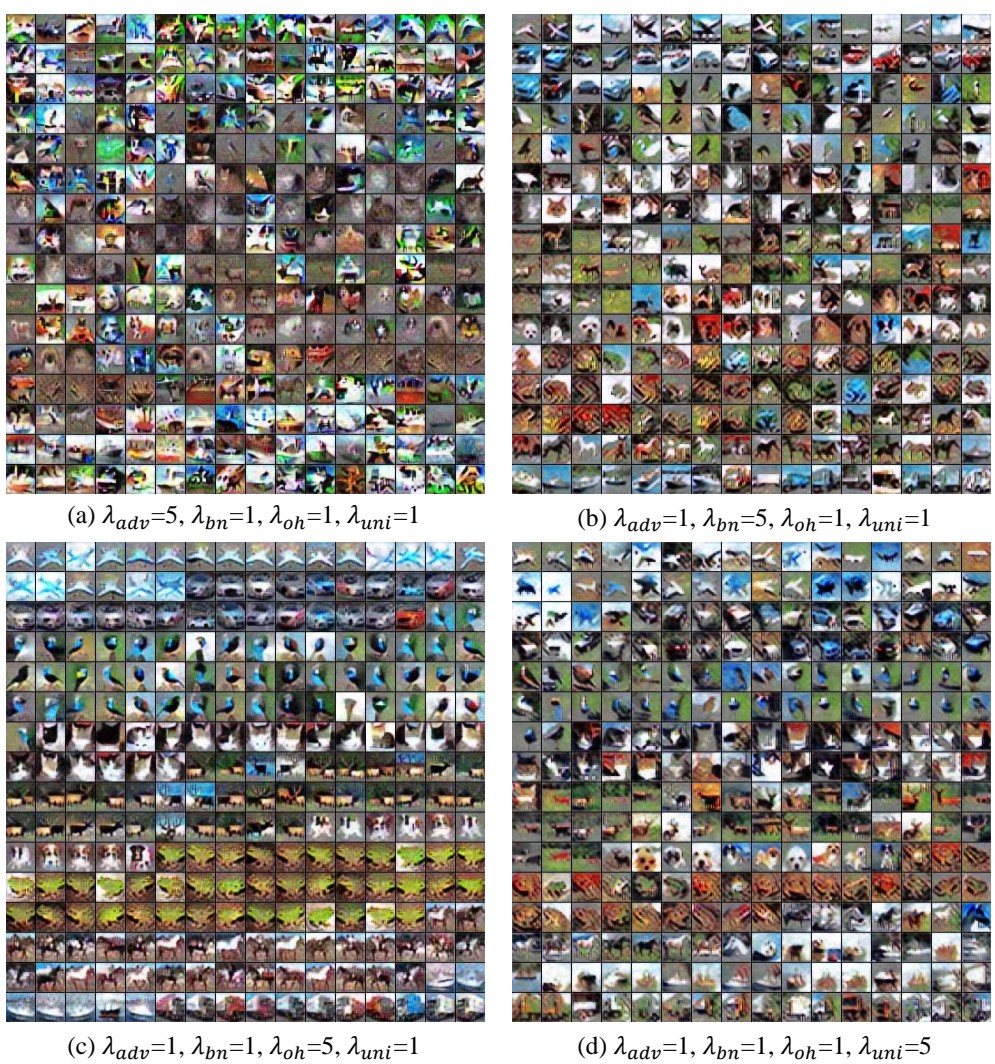

(a) $\lambda_{adv}$=5, $\lambda_{bn}$=1, $\lambda_{oh}$=1, $\lambda_{uni}$=1

(b) $\lambda_{adv}$=1, $\lambda_{bn}$=5, $\lambda_{oh}$=1, $\lambda_{uni}$=1

(c) $\lambda_{adv}$=1, $\lambda_{bn}$=1, $\lambda_{oh}$=5, $\lambda_{uni}$=1

(d) $\lambda_{adv}$=1, $\lambda_{bn}$=1, $\lambda_{oh}$=1, $\lambda_{uni}$=5

Figure 8: Synthetic samples under different paramters.

From the results, we find that $\mathcal{L}_{adv}$ significantly improves the average robustness under all adversarial attacks, especially against FGSM and PGD. From the synthetic samples, we find that $\mathcal{L}_{bn}$ helps to improve the sample quality, as shown in Fig. 8 (b), making the student more applicable to the

Table 8: Result of average robustness(%) (Avg. ↑), worst-class(%) (Worst ↑), and normalized standard deviation (NSD ↓) on different $\lambda_{adv}$, $\lambda_{bn}$, $\lambda_{oh}$ and $\lambda_{uni}$. The best results are **bolded**, and the second best results are underlined.

| Parameters | | | | Clean | | | FGSM | | | PGD | | | CW | | | AA | | |
|---|---|---|---|---|---|---|---|---|---|---|---|---|---|---|---|---|---|---|
| $\lambda_{adv}$ | $\lambda_{bn}$ | $\lambda_{oh}$ | $\lambda_{uni}$ | Avg. | Worst | NSD | Avg. | Worst | NSD | Avg. | Worst | NSD | Avg. | Worst | NSD | Avg. | Worst | NSD |
| 1 | 1 | 1 | 1 | 81.23 | 67.60 | 0.100 | 50.59 | 31.40 | 0.258 | 41.55 | 16.90 | 0.359 | 41.30 | 18.90 | 0.337 | 39.73 | 16.90 | 0.350 |
| 5 | 1 | 1 | 1 | **81.32** | **70.00** | **0.089** | **53.26** | **34.40** | 0.229 | **44.23** | **20.80** | 0.316 | 41.93 | 20.20 | 0.301 | 39.64 | 18.20 | 0.328 |
| 1 | 5 | 1 | 1 | 80.10 | 66.40 | **0.089** | 50.75 | 31.20 | **0.204** | 41.16 | 18.50 | **0.315** | 40.94 | 20.50 | **0.291** | 39.06 | **19.30** | **0.296** |
| 1 | 1 | 5 | 1 | 72.97 | 52.20 | 0.153 | 40.65 | 18.30 | 0.318 | 35.07 | 11.10 | 0.385 | 34.74 | 12.60 | 0.363 | 33.67 | 11.80 | 0.383 |
| 1 | 1 | 1 | 5 | 80.55 | 66.70 | 0.107 | 52.55 | 30.10 | 0.265 | 43.47 | 18.50 | 0.366 | **43.19** | 21.10 | 0.335 | **40.74** | 18.50 | 0.364 |
| 1 | 3 | 1 | 3 | 80.51 | 65.20 | 0.110 | 50.97 | 30.00 | **0.235** | 41.84 | 19.50 | 0.338 | 41.67 | 17.40 | 0.349 | 39.97 | 19.20 | 0.359 |
| 1 | 5 | 1 | 5 | 79.26 | **65.90** | **0.105** | **51.09** | 31.30 | 0.237 | 42.24 | 20.40 | **0.331** | 41.73 | **19.30** | 0.339 | 40.12 | **20.80** | **0.325** |
| 1 | 7 | 1 | 7 | 79.16 | 64.30 | 0.124 | 50.45 | **31.90** | 0.246 | 41.77 | **21.40** | 0.346 | 41.26 | 18.50 | 0.348 | 39.73 | 20.30 | 0.353 |
| 1 | 9 | 1 | 9 | 78.81 | 65.40 | 0.125 | 50.12 | 30.20 | 0.246 | 41.12 | 18.80 | 0.347 | 40.22 | 18.40 | 0.342 | 38.90 | 18.90 | 0.344 |

Table 9: Robustness of DFKD and corresponding DFRD methods. RN-18 and MN-V2 are abbreviations of ResNet-18 and MobileNet-V2 respectively. The value in parentheses is the change value after transforming into DFRD.

| Student | Method | Clean | FGSM | PGD-20 | $CW_\infty$ | AA |
|---|---|---|---|---|---|---|
| RN-18 | ZSKT | **66.45** | 29.60 | **23.03** | 17.46 | 16.43 |
| | ZSKT+RSLAD | 65.48 (–0.97) | **29.80 (+0.20)** | 22.88 (–0.15) | **18.48 (+1.02)** | **17.16 (+0.73)** |
| | CMI | **76.20** | 32.34 | 28.27 | 26.24 | 25.53 |
| | CMI+RSLAD | 68.29 (–7.91) | **33.34 (+1.00)** | **28.94 (+0.67)** | **27.63 (+1.39)** | **26.85 (+1.32)** |
| | DeepInv | **74.20** | 34.98 | 27.50 | 23.39 | 22.84 |
| | DeepInv+RSLAD | 71.22 (–2.98) | **38.60 (+3.62)** | **27.95 (+0.45)** | **27.23 (+3.84)** | **25.19 (+2.35)** |
| | Fast | **78.51** | 41.04 | 37.23 | 30.31 | 27.95 |
| | Fast+RSLAD | 72.20 (–6.31) | **43.96 (+1.98)** | **39.64 (+2.41)** | **33.96 (+3.65)** | **33.24 (+5.29)** |
| MN-V2 | ZSKT | 52.64 | 22.35 | 19.76 | 15.78 | 14.41 |
| | ZSKT+RSLAD | **54.69 (+2.05)** | **23.62 (+1.27)** | **19.83 (+0.07)** | **15.86 (+0.08)** | **14.85 (+0.44)** |
| | CMI | **67.95** | **35.43** | 17.55 | 15.69 | 6.87 |
| | CMI+RSLAD | 60.70 (–7.25) | 27.90 (–7.53) | **19.70 (+2.15)** | **19.57 (+3.88)** | **18.09 (+11.22)** |
| | DeepInv | **66.94** | **34.98** | 15.26 | 14.97 | 6.19 |
| | DeepInv+RSLAD | 62.77 (–4.17) | 31.66 (–3.32) | **19.13 (+3.87)** | **19.01 (+4.04)** | **15.39 (+9.2)** |
| | Fast | **69.81** | **30.17** | 24.03 | 16.69 | 14.59 |
| | Fast+RSLAD | 58.60 (–11.21) | 29.91 (–0.26) | **24.28 (+0.25)** | **19.46 (+2.77)** | **18.35 (+3.76)** |

actual scenario. We also observe that appropriate $\mathcal{L}_{oh}$ supervision is necessary, but too high will suppress sample diversity and lead to performance degradation. For $\mathcal{L}_{uni}$, it helps to improve the robustness performance under the strongest attack (such as AA) and alleviate the robust unfairness problem. Here, considering the practicality and robust fairness of the model, we focus on $\lambda_{bn}$ and $\lambda_{uni}$ as parameters tuning. From the bottom four rows of the Tab. 8, we find that it has the best performance in most cases when $\lambda_{bn}$ and $\lambda_{uni}$ are both equal to 5, so we set the hyperparameter $\lambda_{adv}$=1, $\lambda_{bn}$=5, $\lambda_{oh}$=1, $\lambda_{uni}$=5 finally.

# G    TRANSFORMING DFKD INTO DFRD

For most Data-Free Knowledge Distillation (DFKD), they do not involve robustness, in which case we transform them into Data-Free Robust Distillation (DFRD) by adding adversarial noise to synthetic samples and then take synthetic samples and adversarial examples for robust distillation. Here, we use PGD to generate adversarial examples and then apply the same distillation training loss as RSLAD (Zi et al., 2021) to transform them into DFRD. Here, we set step size to 2/255 and attack iteration to 10. The results are shown in Tab. 9.

We observe that incorporating RSLAD into the distillation leads to a slight decrease in clean accuracy, but it achieves a notable improvement in robustness, particularly under the stronger attacks.

This trade-off is consistent with findings in prior adversarial training (Zhang et al., 2019). Therefore, we think that this transformation approach is reasonable and effective.

## H  ADDITIONAL ABLATION STUDY

### H.1  INTERMEDIATE LAYER $l$

Table 10: Ablation study of different Intermediate Layer $l$.

| Layer | Clean | | FGSM | | PGD-20 | |
|---|---|---|---|---|---|---|
| | Avg. | Worst | Avg. | Worst | Avg. | Worst |
| block1 | 78.50 | 64.80 | 48.83 | 30.50 | 40.48 | 20.00 |
| block2 | 78.91 | 64.30 | 49.37 | 29.50 | 41.16 | 18.80 |
| block3 | **79.26** | **65.90** | **51.09** | **31.30** | **42.24** | **23.50** |

In FAEs generation process, we distill non-robust features from the output in the intermediate layer $l$ of the teacher and impose uniform constraints, ensuring that the synthetic samples have an equal tendency to different targets when generating adversarial samples. In this section, we research the sensitivity of FERD's robustness performance to different layers. We distill non-robust features from the outputs of the penultimate, second and third residual modules, respectively. The results are shown in Tab. 10.

We observe that the penultimate residual module performs best in all aspects. Both the robustness and the fairness have achieved the best results. This further demonstrates that the higher layer contains high-level feature information, which is effective for distilling non-robust feature.

### H.2  DISTILLATION EPOCH $e$

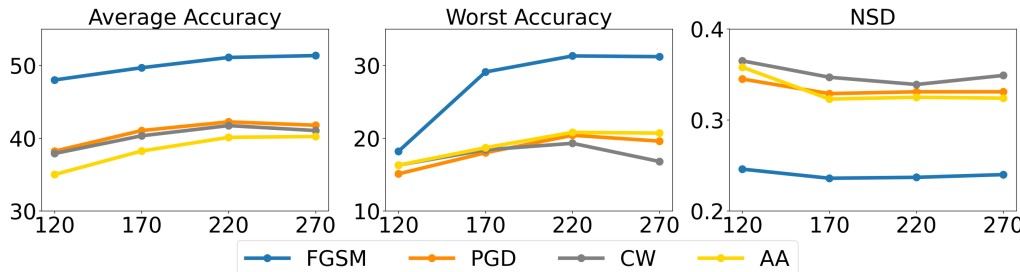

Figure 9: Metric values under different epochs. The horizontal axis is epoch and the vertical axis is metric value.

To explore the impact of the epoch on the student performance, we evaluate the distilled student under different epoch settings. The results are shown in Fig. 9.

With the increase of the epoch, both the average accuracy and the worst-class robustness of the student exhibit a generally increasing trend, reaching a peak at round 220 and then floating slightly, indicating that the robustness is optimal at this time. At the same time, the NSD remains almost constant after 170 epochs. Therefore, it is reasonable to select 220 as the final distillation epochs.

## I  SYNTHETIC SAMPLES FROM DIFFERENT DFRD

This section shows samples synthesized by different DFRD methods. The corresponding synthetic samples are shown in Fig. 10. From the visual results, it is evident that our FERD generates samples that are both more realistic and more diverse, proving that the method is suitable for practical scenarios.

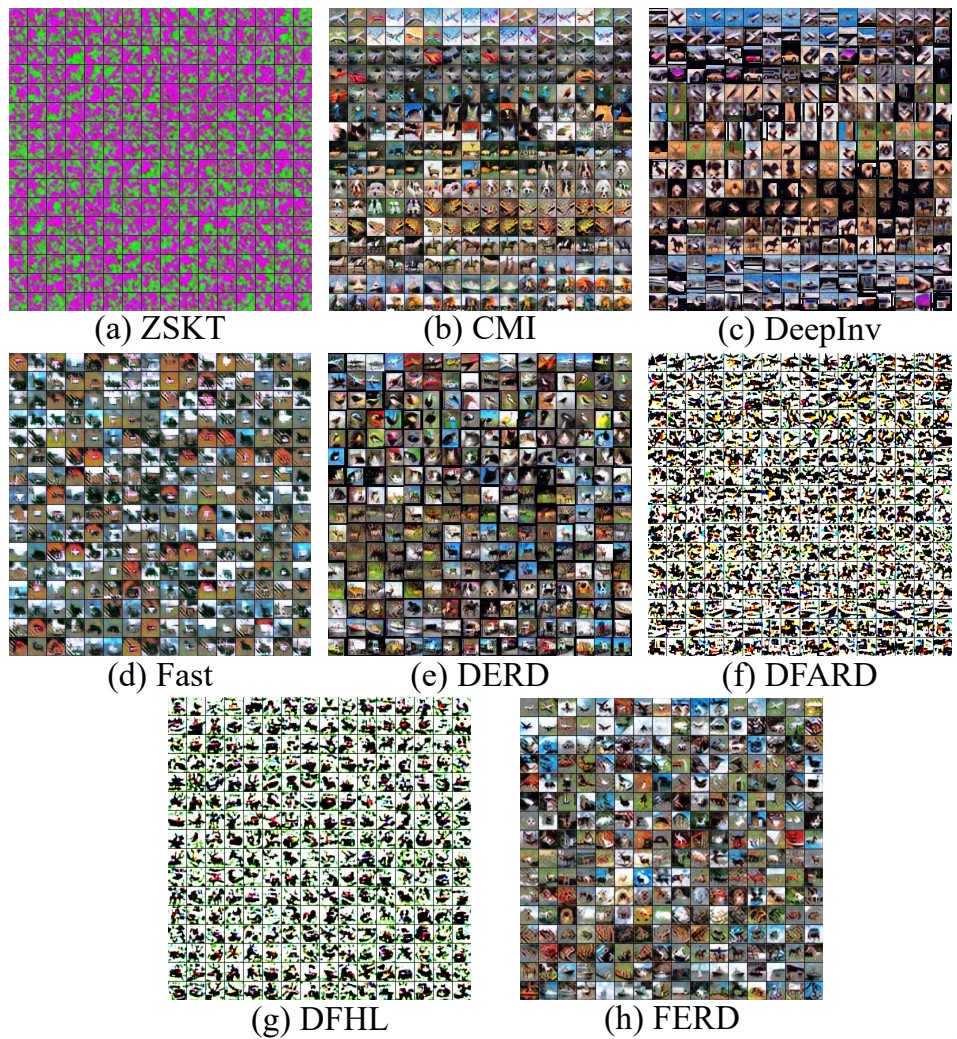

Figure 10: Inverted samples from different DFRD.

## J    LIMITATIONS

Although FERD mitigates the robust fairness by introducing a robust-guided class reweighting strategy, the class imbalance may still affect the final results in some extreme cases. For example, if some classes are very rare in the original data, the problem of unfairness of robustness may not be completely solved by just adjusting the sample proportion. Moreover, the generator needs to optimize to multiple loss functions to synthesize high-quality samples, while the student also needs to perform robust distillation. These processes involve complex optimizations and a large number of iterations, which lead to high computational cost.

## K    THE USE OF LARGE LANGUAGE MODELS

In the writing process for this paper, we employed a large language model for text refinement and grammatical corrections to enhance overall readability and precision. We clarify that the model's role was strictly advisory and confined to language aspects. The background of the study, the design of the experiments, and the analysis of all results represent the independent work of the authors.

