# OpenReview forum: "FERD: Fairness-Enhanced Data-Free Adversarial Robustness Distillation"
_ICLR.cc/2026/Conference — ICLR 2026 Poster_

### Official Review · Reviewer_VYHq · 2025-10-21

**Soundness:** 3
**Presentation:** 3
**Contribution:** 3
**Rating:** 8
**Confidence:** 4

**Summary:**

This paper proposes a new framework, FERD, to address the robustness and fairness issues in data-free adversarial robustness distillation. The authors find that, even with uniformly sampled synthetic data, the robustness of the student model in traditional DFRD methods still varies significantly across categories, and the success rate of adversarial attacks also varies depending on the target class. FERD addresses this issue through two strategies: Robustness-guided class reweighting: This increases the generation of samples for weakly robust classes; Fairness-aware sample and uniform target adversarial sample generation: This improves the class coverage of adversarial samples by uniformly constraining feature levels and uniformizing attack targets.

**Strengths:**

1. This paper is easy to follow.

2. The proposed method is novel. A class reweighting strategy is proposed to enhance robustness against weak classes. FAEs and UTAEs are designed to improve fairness at both the feature and attack target levels.

3. The experimental result demonstrates the proposed method can effectively alleviate fairness issues in data-free ARD.

4. This paper also provides an explanation from a theoretical level.

**Weaknesses:**

1. This paper only selected one teacher model for one data set. I am curious whether the method is effective when different teacher models are selected.

2. The experimental result in newer architectures, such as VIT, can further verify the effectiveness.

**Questions:**

See the weakness

---

> ### Author Response · Authors · 2025-11-24
>
> We thank the reviewer for the insightful comments. We provide detailed responses below. We will add this content in the revision.
>
> **Q1: Whether the method is effective when different teacher models are selected?**
>
> A: Thanks for your suggestion. We choose WRN-34-20 trained on CIFAR10 as the teacher model, and the model performance is as follows:
>
> | Teacher Model | Clean | FGSM | PGD | CW | AA |
> |:---:|:---:|:---:|:---:|:---:|:---:|
> | WRN-34-20 | 86.10 | 71.81 | 69.31 | 57.02 | 58.09 |
>
> We choose ResNet18 as the student model, and the experimental results on CIFAR10 are as follows:
>
> | Method | Clean Avg.↑ | Clean Worst↑ | Clean NSD↓ | FGSM Avg.↑ | FGSM Worst↑ | FGSM NSD↓ | PGD Avg.↑ | PGD Worst↑ | PGD NSD↓ | CW Avg.↑ | CW Worst↑ | CW NSD↓ | AA Avg.↑ | AA Worst↑ | AA NSD↓ |
> |:---:|:---:|:---:|:---:|:---:|:---:|:---:|:---:|:---:|:---:|:---:|:---:|:---:|:---:|:---:|:---:|
> | ZSKT | 50.14 | 14.20 | 0.358 | 29.99 | 11.30 | 0.309 | 29.47 | 9.50 | 0.392 | 25.76 | 10.80 | 0.298 | 3.39 | 0.20 | 1.034 |
> | CMI | 73.26 | 57.90 | 0.147 | 51.25 | 38.50 | 0.185 | 43.94 | 29.40 | 0.221 | 41.71 | 29.30 | 0.221 | 26.44 | 9.80 | 0.452 |
> | Deepinv | 69.47 | 56.20 | 0.124 | 51.35 | 40.80 | 0.182 | 42.66 | 31.50 | **0.163** | 43.92 | 30.70 | 0.234 | 24.80 | 13.40 | 0.370 |
> | Fast | 68.95 | 53.30 | 0.136 | 53.21 | 41.70 | 0.183 | 51.86 | 35.70 | 0.202 | 40.15 | 28.50 | 0.233 | 30.28 | 14.70 | 0.395 |
> | DFHL | 70.89 | 52.60 | 0.137 | 52.98 | 39.80 | 0.186 | 51.28 | 36.10 | 0.181 | 43.48 | 28.90 | 0.220 | 36.59 | 15.60 | 0.379 |
> | FERD(Ours) | **78.23** | **61.70** | **0.118** | **60.01** | **42.30** | **0.178** | **54.27** | **36.30** | 0.202 | **45.90** | **30.90** | **0.216** | **38.67** | **16.00** | **0.368** |
>
> As shown in the table, FERD still outperforms all baseline methods when distilled from the WRN-34-20 teacher. Specifically, FERD achieves the highest average robustness and worst-robustness. Under the four attacks, FERD achieves the highest improvement of 6.8%, 2.08% and 0.011 on the three metrics respectively. This is attributed to the transferability of our designed loss functions in Section 4.3, which can invert high-quality images from teacher models with different architectures.
>
> **Q2: The experimental results on newer architectures, such as ViT, can further verify the effectiveness.**
>
> A: We appreciate this suggestion. We are training the code on modern architectures and will update the results soon.

---

> > ### Author Response · Authors · 2025-12-03
> >
> > **Q2: The experimental results on newer architectures, such as ViT, can further verify the effectiveness.**
> >
> > A: We appreciate your suggestion. We choose a robust XCiT-S12 model trained on Imagenet as the teacher, and ResNet18 as student. We choose Fast as the baseline, which is specially designed for large datasets like ImageNet. The results are as follows:
> >
> > | Method | Clean Avg.↑ | Clean NSD↓ | FGSM Avg.↑ | FGSM NSD↓ | PGD Avg.↑ | PGD NSD↓ |
> > |:---:|:---:|:---:|:---:|:---:|:---:|:---:|
> > | Fast | 25.82 | 0.699 | 3.44 | 1.740 | 3.07 | 1.823 |
> > | FERD | **27.20** | **0.588** | **4.50** | **1.299** | **4.10** | **1.613** |
> >
> > As shown in the table, FERD outperforms Fast on ImageNet. This results from the transferable loss functions we designed, which support high-quality inversion across different teacher architectures.

---

### Official Review · Reviewer_jJ8d · 2025-10-23

**Soundness:** 3
**Presentation:** 2
**Contribution:** 3
**Rating:** 4
**Confidence:** 4

**Summary:**

This paper identifies that data-free robust distillation (DFRD) methods create models with unfair robustness, meaning they are robust for some class but non-robust for others. The authors find this is because DFRD trains on an equal number of samples for all classes and uses attacks that are not diverse. They propose FERD, which fixes this by generating more synthetic data for the non-robust classes and creating special adversarial attacks that target all classes uniformly. Experiments show this method improves the robustness of the weakest classes, leading to better overall fairness in the context of DFRD.

**Strengths:**

1. The paper is the first to investigate and address the problem of robust fairness specifically within the context of Data-Free Robustness Distillation (DFRD), an area where fairness considerations have been largely overlooked.
2. The proposed class reweighting strategy introduces a mechanism to generate intentionally imbalanced synthetic data tailored for fairness improvement, adapting generation proportions based on class vulnerability within the data-free constraint.

**Weaknesses:**

1. The paper claims state-of-the-art robust fairness in DFRD but fails to compare against a crucial baseline: combining a standard DFRD data generation method with established fairness-aware adversarial distillation techniques like Fair-ARD or ABSLD. It is plausible that simply applying existing fairness distillation losses to student training, using data from a generic DFRD generator, could yield comparable fairness improvements. Without this comparison, the paper does not demonstrate that FERD's specific components offer benefits beyond a straightforward integration of known DFRD generation and fairness AD methods.

2. Robust fairness methods (e.g., ABSLD) demonstrate their effectiveness by showing significant improvements in worst-case robustness or NSD while maintaining comparable or slightly improved average robustness relative to strong baselines. This isolates the contribution specifically to fairness. However, FERD's results show a large increase in average robustness alongside the worst-case improvement (e.g., Table 1, RN-18/CIFAR-10: Avg. AA +3% vs. Worst AA +1% compared to DFHL). This large average gain makes it difficult to determine if the worst-case improvement is a direct outcome of the fairness mechanisms or simply a byproduct of the model becoming significantly stronger overall. While NSD improves, the paper does not isolate the fairness contribution by comparing against a baseline with similar average robustness, thus obscuring whether FERD primarily enhances fairness or just general robustness, which happens to also lift the worst-case performance.

3. As a pioneering work in DFRD fairness, the paper's evaluation could be strengthened by providing additional context. Including teacher's own fairness profile and student performance under data-available adversarial distillation—using both standard (e.g., RSLAD) and fairness-aware (e.g., ABSLD) methods—would offer valuable reference optimal points. Without these comparisons, and alongside the significant average robustness gains (Weakness 2), it remains somewhat challenging to fully gauge the significance of FERD's fairness improvements specifically within the constraints of the data-free setting.

**Questions:**

-	While Section 2.1 discusses several DFRD methods (DFARD, DERD, DFHL), your experiments only include DFHL as a direct DFRD baseline, primarily comparing against adapted DFKD methods. Could you explain the rationale for omitting DFARD and DERD from the empirical comparison and for this specific choice of baselines?
-	For the adapted DFKD baselines, why did you use standard PGD for attack generation instead of the inner maximization objective proposed in the original RSLAD paper? Could this choice underestimate the adapted baselines' potential robustness?

This paper tackles the important problem of DFRD fairness with strong results. However, concerns about baselines and context limit the current evaluation. I am open to revising my score based on the rebuttal.

---

> ### Author Response · Authors · 2025-11-24
>
> We thank the reviewer for the insightful comments. We provide detailed responses below. We will add this content in the revision.
>
> **W1: The paper fails to compare against a crucial baseline: combining a standard DFRD data generation method with established fairness-aware adversarial distillation techniques like Fair-ARD or ABSLD.**
>
> A: Thanks for your suggestion. As requested, we have combined standard DFRD data generation methods with Fair-ARD. The experimental results are as follows:
>
> | Method | Clean Avg.↑ | Clean Worst↑ | Clean NSD↓ | FGSM Avg.↑ | FGSM Worst↑ | FGSM NSD↓ | PGD Avg.↑ | PGD Worst↑ | PGD NSD↓ |
> |:---:|:---:|:---:|:---:|:---:|:---:|:---:|:---:|:---:|:---:|
> | CMI | 68.29 | 51.20 | 0.170 | 46.73 | 36.40 | 0.199 | 43.07 | 32.50 | 0.227 |
> | CMI+Fair-ARD | 72.08 | 56.80 | 0.149 | 52.31 | 40.40 | 0.177 | 48.18 | 33.50 | 0.196 |
> | DeepInv | 71.22 | 58.90 | 0.130 | 51.34 | 36.90 | 0.196 | 42.22 | 29.00 | 0.221 | 42.11 | 28.10 | 0.198 | 25.19 | 11.60 | 0.445 |
> | DeepInv+Fair-ARD | 72.24 | 59.90 | 0.139 | 49.50 | 38.90 | 0.165 | 43.06 | 30.30 | **0.186**  |
> | FERD(Ours) | **79.26** | **65.90** | **0.110** | **61.04** | **46.10** | **0.160** | **54.93** | **39.80** | 0.197 |
>
> We find that Fair-ARD indeed enhances robust fairness, achieving better results in worst-robustness and NSD metrics. For instance, CMI+Fair-ARD improves FGSM worst-robustness by 4.0% and reduces NSD by 0.022 compared to vanilla CMI. However, our method still achieves the best results in most situations. Under the FGSM and PGD attacks, the worst robustness of FERD is 5.7% and 6.3% higher than that of the optimal baseline. This is because the quality of the synthetic images from the baselines is poor, and their distance from the decision boundary cannot truly reflect the strength of robustness, so Fair-ARD yields limited improvement.
>
> **W2: FERD's results show a large increase in average robustness. This large average gain makes it difficult to determine if the worst-case improvement is a direct outcome of the fairness mechanisms or simply a byproduct of the model becoming significantly stronger overall.**
>
> A: Thanks for your insightful comments on the experimental results. In fact, the primary goal of our method is to enhance robust fairness, and only secondarily will we aim to improve average robustness. To address your concerns, we have designed the following experiment. We observe that most baselines show robustness to FGSM attack between 50% and 55%. To avoid the impact of a large gain in average robustness, we stop distillation when robustness reaches 55% and observe the robust fairness of the model. The performance is as follows:
> | Method | Clean Avg.↑ | Clean Worst↑ | Clean NSD↓ | FGSM Avg.↑ | FGSM Worst↑ | FGSM NSD↓ | PGD Avg.↑ | PGD Worst↑ | PGD NSD↓ | CW Avg.↑ | CW Worst↑ | CW NSD↓ | AA Avg.↑ | AA Worst↑ | AA NSD↓ |
> |:---:|:---:|:---:|:---:|:---:|:---:|:---:|:---:|:---:|:---:|:---:|:---:|:---:|:---:|:---:|:---:|
> | ZSKT | 65.48 | 32.20 | 0.280 | 44.17 | 26.00 | 0.324 | 43.06 | 23.50 | 0.349 | 32.02 | 21.50 | 0.256 | 17.16 | 2.20 | 0.780 |
> | CMI | 68.29 | 51.20 | 0.170 | 46.73 | 36.40 | 0.199 | 43.07 | 32.50 | 0.227 | 36.70 | 28.90 | 0.209 | 26.87 | 14.60 | 0.441 |
> | DeepInv | 71.22 | 58.90 | 0.130 | 51.34 | 36.90 | 0.196 | 42.22 | 29.00 | 0.221 | 42.11 | 28.10 | 0.198 | 25.19 | 11.60 | 0.445 |
> | Fast | 72.20 | 57.80 | 0.139 | **56.23** | **43.80** | 0.188 | **53.70** | 37.00 | 0.201 | 41.42 | 28.50 | 0.228 | 33.19 | 17.60 | 0.422 |
> | DFHL | 74.42 | 58.60 | 0.136 | 53.56 | 41.10 | 0.176 | 51.23 | 36.90 | 0.198 | **42.15** | 26.90 | 0.214 | **36.37** | 18.50 | 0.369 |
> | FERD(Ours) | **75.18** | **63.80** | **0.105** | 55.16 | 42.80 | **0.116** | 49.80 | **39.10** | **0.142** | 41.93 | **32.40** | **0.146** | 33.01 | **18.90** | **0.291** |
>
> From the table we can find that, even when the average robustness is not high or even lower than the baselines, FERD's results in the robust fairness metrics still exceed the baseline, especially in NSD metric. Under the four attacks, the performance of FERD in NSD are 0.06, 0.056, 0.052 and 0.078 respectively lower than the best baseline. This is attributed to the fact that our reweighting sampling strategy generates more samples from the weak-robust categories to promote fairness. This demonstrates that robust fairness is the primary objective during our distillation process. What's more, we believe that significantly enhancing the average robustness is also an important contribution.

---

> > ### Author Response · Authors · 2025-11-24
> >
> > **W3: The paper's evaluation could be strengthened by providing additional context. Including teacher's own fairness profile and student performance under data-available adversarial distillation.**
> >
> > A: We thank the reviewer for the suggestion.
> > **For the teacher's own fairness profile**, we provide detailed information about the teacher as follows:
> > | Dataset | Model |  Clean Avg.↑ | Clean Worst↑ | Clean NSD↓ | FGSM Avg.↑ | FGSM Worst↑ | FGSM NSD↓ | PGD Avg.↑ | PGD Worst↑ | PGD NSD↓ | CW Avg.↑ | CW Worst↑ | CW NSD↓ | AA Avg.↑ | AA Worst↑ | AA NSD↓ |
> > | :---: | :---: | :---: | :---: | :---: | :---: | :---: | :---: | :---: | :---: | :---: | :---: | :---: | :---: | :---: | :---: | :---: |
> > | CIFAR10 | WRN-34-10 | 86.55 | 73.40 | 0.093 | 71.85 | 53.80 | 0.153 | 69.84 | 51.00 | 0.171 | 56.23 | 34.90 | 0.243 | 57.30 | 31.00 | 0.295 |
> > | CIFAR100 | WRN-34-10 | 64.32 | 37.60 | 0.231 | 48.50 | 20.80 | 0.345 | 46.41 | 18.30 | 0.369 | 37.40 | 16.10 | 0.423 | 31.15 | 5.20 | 0.614 |
> > | Tiny-ImageNet | PreActResNet34 | 48.96 | 20.30 | 0.359 | 34.64 | 12.20 | 0.449 | 32.81 | 11.30 | 0.462 | 26.12 | 9.90 | 0.484 | 18.56 | 1.00 | 0.835 |
> >
> > The impact of robust fairness of the teacher on the student is discussed in Section 3.1 of the paper. The following is the detailed data under AA attack:
> >
> > | Class | Teacher | ZSKT | CMI | DeepInv | Fast | DFHL | FERD(Ours) |
> > |:---:|:---:|:---:|:---:|:---:|:---:|:---:|:---:|
> > | 0 | 67.90 | 27.50 | 36.10 | 42.60 | 42.60 | 41.20 | 55.10 |
> > | 1 | 79.30 | 34.00 | 43.90 | 39.50 | 35.50 | 37.50 | 57.40 |
> > | 2 | 45.80 | 4.70 | 14.60 | 16.30 | 17.60 | 21.10 | 30.60 |
> > | 3 | 31.00 | 4.60 | 17.80 | 11.60 | 18.00 | 18.50 | 20.90 |
> > | 4 | 37.20 | 4.90 | 17.00 | 12.70 | 19.30 | 22.10 | 24.30 |
> > | 5 | 43.70 | 9.80 | 17.30 | 19.20 | 22.40 | 31.60 | 32.30 |
> > | 6 | 58.10 | 2.20 | 17.40 | 19.90 | 35.40 | 43.30 | 33.70 |
> > | 7 | 64.70 | 30.50 | 26.50 | 34.90 | 35.10 | 37.10 | 42.10 |
> > | 8 | 76.90 | 34.40 | 45.40 | 31.80 | 60.00 | 59.00 | 57.10 |
> > | 9 | 68.40 | 19.00 | 32.70 | 23.40 | 46.00 | 52.40 | 48.00 |
> > | MIN | 31.00 | 2.20 | 14.60 | 11.60 | 17.60 | 18.50 | **20.90** |
> > | AVE | 57.30 | 17.16 | 26.87 | 25.19 | 33.19 | 36.38 | **40.15** |
> > | NSD | 0.295 | 0.780 | 0.441 | 0.445 | 0.422 | 0.369 | **0.341** |
> >
> > We compare the classification performance of the teacher and the students distilled by six methods on different categories. We find students performance is consistent with the trend of teacher performance. This is attributed to the guidance of the teacher in the distillation. However, the robust fairness of the students will become worse. For instance, ZSKT student scores 0.485 higher than teachers on the NSD metric, and the accuracy of the worst class is only 2.20%. Nonetheless, FERD outperforms the five baselines in both Min Acc, Max Acc, and NSD metrics.
> >
> > **For data-available adversarial distillation**, we transfer available components into it. Since there is no generation stage, we incorporate the reweighting sampling strategy and UTAEs into the data-available distillation, and we compare with the data-free setting. The experimental results are as follows:
> >
> > | Student | Data | Clean Avg.↑ | Clean Worst↑ | Clean NSD↓ | FGSM Avg.↑ | FGSM Worst↑ | FGSM NSD↓ | PGD Avg.↑ | PGD Worst↑ | PGD NSD↓ |
> > |:---:|:---:|:---:|:---:|:---:|:---:|:---:|:---:|:---:|:---:|:---:|
> > | RN18 | Free | 79.26 | 65.90 | 0.110 | 61.04 | 46.10 | 0.160 | 54.93 | 39.80 | 0.197 |
> > | RN18 | Available | **86.84** | **74.40** | **0.082** | **68.13** | **49.90** | **0.149** | **61.94** | **42.50** | **0.192** |
> > | MN-V2 | Free | 78.34 | 64.10 | 0.114 | 62.08 | 49.90 | 0.143 | 53.08 | 38.20 | 0.199 |
> > | MN-V2 | Available | **86.01** | **73.90** | **0.084** | **67.28** | **50.20** | **0.138** | **60.14** | **40.40** | **0.192** |
> >
> > From the table we observe that the performance of FERD is overall better in data-available distillation. It is reasonable because DFKD requires inverting the original data. This proves that FERD remains effective in data-available distillation. It is noteworthy that the gaps in robust fairness metrics are smaller than the gaps in average robustness. For instance, the worst-robustness gap of MN-V2 student under FGSM attack is 0.03%, but the average robustness gap is 5.20%. This indicates that FERD focuses more attention on improving the performance of weak-robust categories.

---

> > > ### Author Response · Authors · 2025-11-24
> > >
> > > **Q1: Could you explain the rationale for omitting DFARD and DERD from the empirical comparison and for this specific choice of baselines?**
> > >
> > > A: We thank the reviewer for pointing this out. When we reproduced DFARD and DERD on CIFAR10, we found that DFHL performed better than them, so we chose to compare with the superior method. To address your concerns, we compare our method with DFARD and DERD. The results are as follows:
> > >
> > > **CIFAR10**
> > > | Student | Method | Clean Avg.↑ | Clean Worst↑ | Clean NSD↓ | FGSM Avg.↑ | FGSM Worst↑ | FGSM NSD↓ | PGD Avg.↑ | PGD Worst↑ | PGD NSD↓ | CW Avg.↑ | CW Worst↑ | CW NSD↓ | AA Avg.↑ | AA Worst↑ | AA NSD↓ |
> > > |:---:|:---:|:---:|:---:|:---:|:---:|:---:|:---:|:---:|:---:|:---:|:---:|:---:|:---:|:---:|:---:|:---:|
> > > | Resnet18 | DERD | 73.91 | 53.70 | 0.158 | 51.75 | 38.70 | 0.199 | 44.36 | 29.60 | 0.225 | 40.06 | 29.10 | 0.201 | 28.60 | 11.80 | 0.444 |
> > > | Resnet18 | DFARD | 67.93 | 45.90 | 0.260 | 53.69 | 34.60 | 0.274 | 52.30 | 33.80 | 0.264 | 43.60 | 29.20 | 0.239 | 34.48 | 12.00 | 0.552 |
> > > | Resnet18 | FERD(Ours) | **79.26** | **65.90** | **0.110** | **61.04** | **46.10** | **0.160** | **54.93** | **39.80** | **0.197** | **47.03** | **31.90** | **0.199** | **40.15** | **20.90** | **0.341** |
> > > | MN-V2 | DERD | 69.80 | 48.70 | 0.175 | 53.15 | 39.00 | 0.186 | 43.39 | 32.20 | 0.203 | 42.07 | 31.00 | 0.190 | 25.80 | 12.00 | 0.404 |
> > > | MN-V2 | DFARD | 66.32 | 40.10 | 0.287 | 51.78 | 29.00 | 0.311 | 48.99 | 27.00 | 0.310 | 42.60 | 31.80 | 0.241 | 31.80 | 8.00 | 0.617 |
> > > | MN-V2 | FERD(Ours) | **78.34** | **64.10** | **0.114** | **62.08** | **49.90** | **0.143** | **53.08** | **38.20** | **0.199** | **48.12** | **35.70** | **0.178** | **38.06** | **20.30** | **0.349** |
> > >
> > > **CIFAR100**
> > > | Student | Method | Clean Avg.↑ | Clean Worst↑ | Clean NSD↓ | FGSM Avg.↑ | FGSM Worst↑ | FGSM NSD↓ | PGD Avg.↑ | PGD Worst↑ | PGD NSD↓ | CW Avg.↑ | CW Worst↑ | CW NSD↓ | AA Avg.↑ | AA Worst↑ | AA NSD↓ |
> > > |:---:|:---:|:---:|:---:|:---:|:---:|:---:|:---:|:---:|:---:|:---:|:---:|:---:|:---:|:---:|:---:|:---:|
> > > | Resnet18 | DERD | 54.01 | 28.10 | 0.295 | 38.02 | 14.70 | 0.429 | 34.92 | 12.10 | 0.464 | 29.57 | 11.00 | 0.493 | 21.05 | 1.40 | 0.856 |
> > > | Resnet18 | DFARD | 50.43 | 21.80 | 0.340 | 36.53 | 14.10 | 0.452 | 34.71 | 12.90 | 0.473 | 29.61 | 10.30 | 0.481 | 20.20 | 1.60 | 0.825 |
> > > | Resnet18 | FERD(Ours) | **55.91** | **29.00** | **0.289** | **39.48** | **15.30** | **0.415** | **36.78** | **13.00** | **0.454** | **39.92** | **11.90** | **0.471** | **21.84** | **1.70** | **0.806** |
> > > | MN-V2 | DERD | 54.12 | 29.10 | 0.291 | 36.21 | 15.20 | 0.423 | 32.68 | 13.10 | 0.464 | 28.78 | 12.40 | 0.469 | 18.91 | 1.70 | 0.831 |
> > > | MN-V2 | DFARD | 49.02 | 20.60 | 0.340 | 35.31 | 13.90 | 0.430 | 33.38 | 10.50 | 0.475 | 29.21 | 12.70 | 0.468 | 19.31 | 1.60 | 0.826 |
> > > | MN-V2 | FERD(Ours) | **56.14** | **29.20** | **0.285** | **40.07** | **17.10** | **0.404** | **35.01** | **13.40** | **0.457** | **31.44** | **13.70** | **0.448** | **21.20** | **2.30** | **0.793** |
> > >
> > > **Tiny-Imagenet**
> > > | Student | Method | Clean Avg.↑ | Clean Worst↑ | Clean NSD↓ | FGSM Avg.↑ | FGSM Worst↑ | FGSM NSD↓ | PGD Avg.↑ | PGD Worst↑ | PGD NSD↓ | CW Avg.↑ | CW Worst↑ | CW NSD↓
> > > |:---:|:---:|:---:|:---:|:---:|:---:|:---:|:---:|:---:|:---:|:---:|:---:|:---:|:---:|
> > > | PreActResNet18 | DERD | 36.32 | 9.40 | 0.470 | 25.13 | 5.70 | 0.499 | 23.74 | 5.60 | 0.505 | 19.28 | 5.60 | 0.513 |
> > > | PreActResNet18 | DFARD | 40.37 | 11.60 | 0.448 | 25.02 | 9.10 | 0.486 | 23.80 | 7.40 | 0.516 | 19.04 | 6.80 | 0.478 |
> > > | PreActResNet18 | FERD(Ours) | **43.88** | **12.30** | **0.439** | **26.84** | **9.20** | **0.465** | **25.51** | **8.00** | **0.483** | **20.51** | **7.20** | **0.463** |
> > >
> > > As shown in results, our method outperforms DERD and DFARD especially on worst-robustness and NSD metrics. For instance, the improvement of FERD is up to 10.9% and 0.103 in worst-robustness and NSD respectively on CIFAR10. This is because the goal of DERD and DFARD is to enhance the overall robustness of the model, while neglecting the issue of robust fairness.

---

> > > > ### Author Response · Authors · 2025-11-24
> > > >
> > > > **Q2: For the adapted DFKD baselines, why did you use standard PGD for attack generation instead of the inner maximization objective proposed in the original RSLAD paper?**
> > > >
> > > > A: Thank you for your suggestion. We attempted to generate adversarial examples by using the inner maximization objective proposed in RSLAD. The results are as follows:
> > > >
> > > > | Method | Clean | FGSM | PGD | CW | AA |
> > > > |:---:|:---:|:---:|:---:|:---:|:---:|
> > > > | CMI+PGD | 68.29 | 46.73 | 43.07 | 36.70 | 26.87 |
> > > > | CMI+RSLAD | 67.20 | 49.73 | 42.40 | 37.31 | 26.53 |
> > > > | DeepInv+PGD | 71.22 | 51.34 | 42.22 | 42.11 | 25.19 |
> > > > | DeepInv+RSLAD | 70.34 | 51.02 | 43.81 | 43.48 | 25.01 |
> > > > | FERD | **79.86** | **61.39** | **54.27** | **46.22** | **39.33** |
> > > >
> > > > The results show that RSLAD's method of generating adversarial examples cannot improve the robustness of the model for all attacks. For instance, the robustness of the two DFKD baselines decreases by 0.34% and 0.18% respectively under AA attack. We speculate the reason is that the synthetic samples are not of high quality. The prediction of the teacher model on these samples cannot truly reflect the label, so the performance of the adversarial examples generated by this method will decrease. However, FERD still significantly outperforms these improved baselines across all metrics. We attribute this performance gap to the superior quality of the synthetic data generated by FERD, which is critical for maximizing the efficacy of adversarial distillation.

---

> > > > > ### Comment · Reviewer_jJ8d · 2025-11-25
> > > > >
> > > > > **Regarding the response to W1:** I appreciate the additional experiments combining standard DFRD generation with Fair-ARD. However, the omission of ABSLD, which represents a more recent in robust fairness than Fair-ARD, weakens the assessment of the proposed method. Furthermore, the authors' observation that the performance gap is due to the "poor quality of synthetic images from the baselines" highlights a methodological flaw in the comparison. Comparing FERD against "CMI/DeepInv + Fair-ARD" conflates the benefits of FERD's generator with the benefits of its fairness mechanisms. This comparison fails to demonstrate that its specific fairness components (Reweighting and UTAEs) are inherently superior to established fairness distillation losses when given data of comparable quality. Additionally, please include the AutoAttack results for these comparisons.
> > > > >
> > > > > **Regarding the response to W2:** While the early stopping experiment demonstrates fairness at lower performance levels, it does not explain the underlying mechanism behind the substantial average robustness gain observed in the main results. Could you provide an analysis or justification for why FERD leads to such a significant boost in overall robustness?
> > > > >
> > > > > **Regarding the Data-Available Experiments:** Thank you for providing the Data-Available results. However, the reported PGD-20 accuracy of 61.94% for ResNet-18 appears anomalously high compared to the consensus in the AD literature, where performance typically falls in the 54-57% range. This discrepancy prevents a meaningful assessment of your method's actual contribution.
> > > > > My original intent in requesting the Data-Available experiment was to rigorously isolate and evaluate the effectiveness of FERD's specific fairness components—Reweighting and UTAEs—independent of the data-free generation process. I strongly request a side-by-side comparison within your exact experimental setting. This comparison should include Standard AD baselines, such as RSLAD or AdaAD, to establish a reliable upper bound for 'Average Robustness' in your environment. More importantly, it must include Fairness-Aware AD baselines, such as Fair-ARD or ABSLD, to directly compare FERD against SOTA fairness AD methods. If these reproduced baselines also achieve nearly 60% PGD accuracy in your setup, it would cast high doubt on your experimental setup. Conversely, if the baselines perform at expected levels (54-56%) while FERD alone reaches 62%, a detailed explanation is required. Without demonstrating that FERD outperforms RSLAD, or AdaAD under the exact same training and evaluation conditions—including AutoAttack—the claim that FERD's fairness mechanisms are effective remains unproven.
> > > > >
> > > > > While this work addresses the intersection of Data-Free Learning, Robust Fairness, and Adversarial Distillation, it fails to benchmark against state-of-the-art techniques in each field like ABSLD or AdaAD, thereby conflating the benefits of a superior data generator with those of the proposed fairness mechanisms. **Since the paper claims contributions in two distinct directions—Data Generation and Fairness-oriented Optimization—it is important to validate the efficacy of each component separately against the relevant SOTA baselines.**
> > > > >
> > > > >
> > > > > Additional concern
> > > > >
> > > > > - There is a significant and concerning discrepancy between the PGD and AutoAttack performance reported in Tables 1 and 2. In standard robust models on CIFAR-10, the gap between PGD-20 and AA typically falls within a small range, rarely exceeding 10%, yet the reported results show discrepancies exceeding 26% in some cases. Such a massive gap is a classic indicator of gradient obfuscation or insufficient attack strength during the PGD evaluation. Furthermore, verification of the evaluation protocol is currently impossible because the submitted code is incomplete; the adversarial accuracy measurement logic exists only as a snippet and appears to be missing in the main training loop. This lack of transparency, combined with the suspicious performance gaps, suggests that the evaluations may not have been conducted correctly.
> > > > >
> > > > >
> > > > >
> > > > > - Minor note: Regarding the submission policy, I found that the submitted source code contains the author's name.

---

> > > > > > ### Author Response · Authors · 2025-11-28
> > > > > >
> > > > > > **Q1: The omission of ABSLD, which represents a more recent method in robust fairness than Fair-ARD, weakens the assessment of the proposed method.**
> > > > > >
> > > > > > Thanks for your suggestion. As requested, we have supplemented the experiment with the addition of ABSLD. The experimental results are as follows:
> > > > > >
> > > > > > | Method | Clean Avg.↑ | Clean Worst↑ | Clean NSD↓ | FGSM Avg.↑ | FGSM Worst↑ | FGSM NSD↓ | PGD Avg.↑ | PGD Worst↑ | PGD NSD↓ | AA Avg.↑ | AA Worst↑ | AA NSD↓ |
> > > > > > |:---:|:---:|:---:|:---:|:---:|:---:|:---:|:---:|:---:|:---:|:---:|:---:|:---:|
> > > > > > | CMI | 68.29 | 51.20 | 0.170 | 33.26 | 19.50 | 0.337 | 30.08 | 16.70 | 0.381 | 26.87 | 14.60 | 0.441 |
> > > > > > | CMI+ABSLD | 76.20 | 58.60 | 0.144 | 46.71 | 27.60 | 0.305 | 41.56 | 22.50 | 0.341 | 33.04 | 15.30 | 0.430 |
> > > > > > | CMI+Fair-ARD | 72.08 | 56.80 | 0.149 | 40.29 | 23.10 | 0.300 | 36.13 | 17.80 | 0.329 | 32.45 | 13.90 | 0.365 |
> > > > > > | DeepInv | 71.22 | 58.90 | 0.130 | 38.57 | 22.40 | 0.304 | 29.10 | 14.60 | 0.359 | 25.19 | 11.60 | 0.445 |
> > > > > > | DeepInv+ABSLD | 76.04 | 53.80 | 0.151 | 50.32 | 30.20 | 0.246 | 44.36 | 24.40 | 0.297 | 34.52 | 13.90 | 0.383 |
> > > > > > | DeepInv+Fair-ARD | 72.24 | 59.90 | 0.139 | 42.34 | 27.90 | 0.262 | 37.23 | 22.50 | 0.300 | 20.91 | 10.20 | 0.420 |
> > > > > > | FERD | **79.26** | **65.90** | **0.110** | **51.08** | **31.30** | **0.237** | **44.68** | **25.30** | **0.290** | **40.15** | **20.90** | **0.341** |
> > > > > >
> > > > > > From the table, we observe that ABSLD indeed outperforms Fair-ARD, improving both robustness and fairness metrics. However, FERD still performs better than them, which is attributed to our high quality synthetic data. Although FERD and baselines are not distilled on fair-quality images, it indeed generates high-quality samples that are not affected by teacher bias. We think this is also a valid innovation point. To fully address your concerns, we use the fair-quality images to conduct experiments in the replies to the following questions.
> > > > > >
> > > > > > **Q2: This comparison fails to demonstrate that its specific fairness components are inherently superior to established fairness distillation losses when given data of comparable quality. And it fails to benchmark against state-of-the-art techniques in Robust Fairness, and Adversarial Distillation.**
> > > > > >
> > > > > > To demonstrate the effectiveness of the fair component, we perform distillation on original CIFAR10 data to compare with RSLAD and ABSLD. The experimental results are as follows:
> > > > > >
> > > > > > | Method | Clean Avg.↑ | Clean Worst↑ | Clean NSD↓ | FGSM Avg.↑ | FGSM Worst↑ | FGSM NSD↓ | PGD Avg.↑ | PGD Worst↑ | PGD NSD↓ | AA Avg.↑ | AA Worst↑ | AA NSD↓ |
> > > > > > |:---:|:---:|:---:|:---:|:---:|:---:|:---:|:---:|:---:|:---:|:---:|:---:|:---:|
> > > > > > | ABSLD | 83.77 | 71.20 | 0.090 | 57.92 | 36.60 | 0.206 | 53.24 | 31.70 | 0.236 | 48.35 | 24.60 | 0.294 |
> > > > > > | RSLAD | **84.79** | 68.70 | 0.106 | **61.50** | 36.80 | 0.234 | **56.12** | 30.30 | 0.267 | **51.88** | 24.50 | 0.315 |
> > > > > > | FERD | 83.24 | **73.40** | **0.076** | 55.99 | **38.70** | **0.189** | 51.04 | **31.90** | **0.235** | 45.59 | **25.10** | **0.285** |
> > > > > >
> > > > > > From the table we find that our fair component has the effect of promoting robust fairness. FERD improves the worst robustness and NSD metrics, but the average robustness decreases to a certain extent. These results are consistent with the conclusion of previous papers.

---

> > > > > > ### Author Response · Authors · 2025-11-28
> > > > > >
> > > > > > **Q3: Could you provide an analysis or justification for why FERD leads to such a significant boost in overall robustness?**
> > > > > >
> > > > > > Thank you for your feedback. We appreciate that the early stopping experiment has clarified the effectiveness of our fairness mechanisms. The large gain in average robustness is a consequence of improvement in synthetic data quality. When the traditional DFKD methods are applied to the robust teachers, because they do not consider the robustness, the synthetic pictures are usually very fuzzy and different from the actual pictures, which leads to the accuracy of each class of the model cannot reach the upper limit, and the average robustness will consequently decrease. However, by extracting and suppressing the non-robust features related to adversarial perturbations, FERD generates higher quality and robust fair samples from the feature level, thereby improving the accuracy of each class of the model, and the average robustness will consequently be improved.
> > > > > >
> > > > > >
> > > > > > **Q4: The reported PGD-20 accuracy of ResNet-18 appears anomalously high.**
> > > > > >
> > > > > > We sincerely appreciate your expert insight regarding the reported PGD-20 accuracy for ResNet-18. Upon rigorous re-examination, we clarify that the reported data-available results are obtained using a prediction-based PGD protocol, inadvertently underestimates PGD attack strength. However, the implementation of AutoAttack code is correct and is still a reliable metric. As observed from AutoAttack, FERD is still better than baselines, which dose not affect the conclusions. We have rerun the experiments and the results are shown in above questions. We promise to rerun the results based on your suggestion and correct them in the paper before the rebuttal ends. The robustness test code has been added to supplementary material.
> > > > > >
> > > > > >
> > > > > > **Q5: Regarding the submission policy, I found that the submitted source code contains the author's name.**
> > > > > >
> > > > > > Thank you for pointing this out. The string in the submitted file path is simply a leftover from a local development environment and does not carry any meaningful identifying information. Its presence was unintentional and has no effect on the method or results. We will remove or anonymize all such paths in the camera-ready version.

---

> > > > > > > ### Author Response · Authors · 2025-12-03
> > > > > > >
> > > > > > > **Q4: The reported PGD-20 accuracy of ResNet-18 appears anomalously high.**
> > > > > > >
> > > > > > > We have conducted the experiments following your suggestions and updated the corresponding results in the revised manuscript. The experimental evidence confirms that FERD still maintains its superiority over other baselines, and the reason is attributed to our effective fairness mechanism and high-quality synthetic samples. Our conclusions and framework remain valid.

---

### Official Review · Reviewer_WX8r · 2025-10-28

**Soundness:** 3
**Presentation:** 2
**Contribution:** 3
**Rating:** 6
**Confidence:** 3

**Summary:**

The work offers a valuable perspective on robust fairness under data-free settings and provides convincing experimental evidence, though the methodological and theoretical originality are limited.

**Strengths:**

The paper addresses the intersection of data-free robustness distillation and robust fairness, which is an emerging and practically important direction for fairness-aware model compression and deployment on edge devices.

The experiments are extensive and demonstrate consistent improvements in worst-class robustness and normalized standard deviation (NSD) across CIFAR-10, CIFAR-100, and Tiny-ImageNet datasets.

Ablation studies, hyperparameter analyses, and visualization of synthetic samples (Fig. 6, Table 3) enhance the reproducibility and credibility of results.

**Weaknesses:**

Although the paper includes proofs of conjectures (Appendix A.1–A.2), they are mostly intuitive restatements of empirical observations and lack rigorous formalism.

The fairness claim relies mainly on NSD and worst-class robustness. Introducing additional fairness metrics would make the evaluation more convincing.

The manuscript is lengthy and includes numerous large figures and dense equations that affect readability.

**Questions:**

Although the paper includes proofs of conjectures (Appendix A.1–A.2), they are mostly intuitive restatements of empirical observations and lack rigorous formalism.

The fairness claim relies mainly on NSD and worst-class robustness. Introducing additional fairness metrics would make the evaluation more convincing.

The manuscript is lengthy and includes numerous large figures and dense equations that affect readability.

---

> ### Author Response · Authors · 2025-11-24
>
> We thank the reviewer for the insightful comments. We provide detailed responses below. We have also updated the manuscript, with the modified contents highlighted in red for your review.
>
> **W1&Q1: Although the paper includes proofs of conjectures (Appendix A.1–A.2), they are mostly intuitive restatements of empirical observations and lack rigorous formalism.**
>
> A: Thanks for your suggestion. We have supplemented the proof details in Appendices A.1 and A.2. Specifically, we introduce Theorem 1 based on the L-smoothness assumption to formally demonstrate that our reweighting sampling strategy achieves a strictly greater expected risk reduction for weak-robust categories compared to uniform sampling in Appendix A.1. We also mathematically derive that our method minimizes an upper bound of the standard adversarial risk. We have updated the manuscript, with the modified contents highlighted in red for your review.
>
> **W2&Q2: The fairness claim relies mainly on NSD and worst-class robustness. Introducing additional fairness metrics would make the evaluation more convincing.**
>
> A: We appreciate your suggestion. We have added two additional metrics: $\text{Maximum Class-wise Discrepancy}$ and $Worst Bndy.$.
>
> $\text{Maximum Class-wise Discrepancy}$ represents the largest discrepancy of the maximum and minimum accuracy for each category. $Worst Bndy.$ measures the worst-case performance degradation caused by adversarial attacks relative to clean accuracy. Their respective calculation formulas are as follows:
> $\text{Maximum Class-wise Discrepancy} = \max_{i \in N} (Acc_{rob}^i) - \min_{i \in N} (Acc_{rob}^i),$
> $Worst Bndy. = \max_{i \in N} ((1 - Acc_{rob}^i) - (1 - Acc_{std}^i)),$
>
> where $N$ denotes the total number of classes, and $Acc_{.}^i$ denotes the accuracy of the $i$-th class. The lower these values are, the fairer the model is.
>
> The experimental results on CIFAR10 under PGD attack are as follows:
> | Method | $\text{Maximum Class-wise Discrepancy}↓$ | $Worst Bndy.↓$ |
> |:---:|:---:|:---:|
> | ZSKT | 41.20 | 31.60 |
> | CMI | 31.20 | 35.30 |
> | DeepInv | 30.10 | 36.50 |
> | Fast | 33.20 | 26.30 |
> | DFHL | 30.70 | 36.10 |
> | FERD | **28.30** | **25.90** |
>
> As shown in the table, FERD achieves the lowest values on both metrics, achieving a 1.8 and 0.4 decrease, respectively, over the best baseline. The effectiveness of $\text{Maximum Class-wise Discrepancy}$ is attributed to the reweighting strategy. It directly increases the sampling weights for weak-robust categories. By forcing the student to train more on these weak categories, it directly enhances the value of $\min(Acc_{rob})$. The effectiveness of $ Worst Bndy. $ is attributed to the FAEs and UTAEs. They improve the robustness of the student against attacks from all directions.
>
> **W3&Q3: The manuscript is lengthy and includes numerous large figures and dense equations that affect readability.**
>
> A: We thank the reviewer for pointing this out. We have carefully revised Section 4.3 by streamlining dense mathematical formulations and condensing the presentation of the loss functions, thereby reducing visual clutter and enhancing the overall flow and clarity of the methodology. The modified parts have been highlighted in red.

---

### Official Review · Reviewer_wkEv · 2025-11-02

**Soundness:** 3
**Presentation:** 4
**Contribution:** 3
**Rating:** 6
**Confidence:** 4

**Summary:**

This paper studies **robust fairness in Data-Free Robustness Distillation (DFRD)** — transferring robustness from a teacher to a student model without original data.

The authors identify two fairness issues:

(1) class-wise robustness disparity under uniformly sampled synthetic data, and

(2) target-dependent vulnerability under adversarial attacks.

They propose **FERD**, which enhances fairness at both the *proportion* and *distribution* levels:

- **Robustness-guided Class Reweighting** generates more synthetic data for weakly robust classes.
- **Fairness-Aware / Uniform-Target Adversarial Examples (FAEs / UTAEs)** apply uniformity constraints on feature and target spaces to prevent biased robustness transfer.

Experiments on **CIFAR-10, CIFAR-100, and Tiny-ImageNet** show that FERD achieves higher worst-class robustness (+15.1% under FGSM) and lower NSD than prior DFRD methods, suggesting improved fairness and stability.

**Strengths:**

- Novel and well-defined problem setup: fairness within DFRD.
- Dual-level solution addressing both sample imbalance and attack-target bias.
- Consistent empirical results across datasets and attack variants.
- Ablation studies verify component contributions.

**Weaknesses:**

- **Limited theoretical rigor:** The design is intuitive but lacks formal analysis — no explicit robustness bounds, convergence arguments, or assumption statements are provided.
- **Restricted evaluation scope:** Experiments are confined to small datasets (CIFAR-10/100, Tiny-ImageNet) and a single perturbation budget (ε = 8/255), leaving its generality under stronger or diverse attacks unexplored.
- **Hyperparameter opacity:** Key parameters (τ, γ, λ) are fixed without justification or sensitivity analysis, creating uncertainty about robustness across configurations.

**Questions:**

- Could you include **ε-sweep results (ε = {4,6,8,10}/255)** to test whether fairness improvements persist under stronger perturbations?
- Have you explored the **stability of results** under different τ, γ, and λ settings?

---

> ### Author Response · Authors · 2025-11-24
>
> We thank the reviewer for the insightful comments. We provide detailed responses below. We have also updated the manuscript, with the modified contents highlighted in red for your review.
>
> **W1: The design is intuitive but lacks formal analysis — no explicit robustness bounds, convergence arguments, or assumption statements are provided.**
>
> A: Thanks for your suggestion. We have strengthened our design by revising Appendix A.1 and Appendix A.2, which now include explicit assumption statements and formal proofs to support our framework. Specifically, Theorem 1 demonstrates that our reweighting strategy strictly enhances the expected risk reduction for vulnerable categories. Furthermore, Theorem 2 proves that minimizing the Uniform-Target Risk effectively minimizes an upper bound of the Standard Adversarial Risk, thereby establishing explicit robustness bounds for our method. We have updated the manuscript, with the modified contents highlighted in red for your review.
>
> **W2&Q1: Could you include ε-sweep results ($\epsilon$ = {4,6,8,10}/255) to test whether fairness improvements persist under stronger perturbations?**
>
> A: We thank the reviewer for the suggestion. The experiments in the paper are all conducted with $\epsilon$ = 8/255. We have added the requested $\epsilon$-sweep experiments with $\epsilon \in \{4, 6, 10\}/255$ on CIFAR-10. The results are as follows:
>
>
> **$\epsilon$ = 4/255**
> | Method | FGSM Avg.↑ | FGSM Worst↑ | FGSM NSD↓ | PGD Avg.↑ | PGD Worst↑ | PGD NSD↓ | CW Avg.↑ | CW Worst↑ | CW NSD↓ | AA Avg.↑ | AA Worst↑ | AA NSD↓ |
> |:---:|:---:|:---:|:---:|:---:|:---:|:---:|:---:|:---:|:---:|:---:|:---:|:---:|
> | ZSKT | 57.34 | 29.50 | 0.294 | 56.82 | 28.80 | 0.296 | 32.05 | 21.50 | 0.256 | 38.08 | 8.30 | 0.502 |
> | CMI | 60.58 | 45.70 | 0.174 | 59.90 | 44.40 | 0.177 | 36.70 | 28.80 | 0.209 | 47.87 | 30.40 | 0.264 |
> | Deepinv | 62.15 | 49.70 | 0.145 | 59.89 | 45.10 | 0.171 | 42.14 | 28.10 | **0.198** | 47.31 | 30.70 | 0.258 |
> | Fast | 66.35 | 50.70 | 0.155 | 66.01 | 50.10 | 0.159 | 41.45 | 28.60 | 0.228 | 53.26 | 35.50 | 0.247 |
> | DFHL | 66.06 | 53.70 | 0.143 | 65.54 | 53.80 | 0.144 | 42.16 | 26.90 | 0.214 | 56.06 | 39.40 | 0.213 |
> | FERD(Ours) | **72.10** | **57.00** | **0.127** | **70.99** | **56.70** | **0.131** | **47.06** | **31.90** | 0.199 | **61.12** | **42.20** | **0.197** |
>
> **$\epsilon$ = 6/255**
>
> | Method | FGSM Avg.↑ | FGSM Worst↑ | FGSM NSD↓ | PGD Avg.↑ | PGD Worst↑ | PGD NSD↓ | CW Avg.↑ | CW Worst↑ | CW NSD↓ | AA Avg.↑ | AA Worst↑ | AA NSD↓ |
> |:---:|:---:|:---:|:---:|:---:|:---:|:---:|:---:|:---:|:---:|:---:|:---:|:---:|
> | ZSKT | 50.48 | 28.40 | 0.290 | 49.74 | 27.80 | 0.304 | 32.02 | 21.50 | 0.256 | 26.07 | 4.00 | 0.625 |
> | CMI | 53.51 | 41.10 | 0.180 | 51.53 | 39.40 | 0.184 | 36.75 | 28.90 | 0.209 | 36.52 | 22.20 | 0.330 |
> | DeepInv | 56.22 | 43.00 | 0.168 | 50.81 | 36.90 | 0.194 | 42.12 | 28.10 | **0.198** | 35.40 | 20.50 | 0.337 |
> | Fast | 61.48 | 47.50 | 0.167 | 60.62 | 45.90 | 0.169 | 41.44 | 28.50 | 0.228 | 43.21 | 25.80 | 0.321 |
> | DFHL | 58.73 | 45.80 | 0.168 | 60.11 | 46.30 | 0.156 | 42.15 | 26.90 | 0.214 | 46.47 | 29.10 | 0.275 |
> | FERD(Ours) | **66.39** | **52.00** | **0.142** | **63.14** | **47.80** | **0.155** | **47.05** | **31.90** | 0.199 | **50.28** | **30.50** | **0.258** |
>
> **$\epsilon$ = 10/255**
>
> | Method | FGSM Avg.↑ | FGSM Worst↑ | FGSM NSD↓ | PGD Avg.↑ | PGD Worst↑ | PGD NSD↓ | CW Avg.↑ | CW Worst↑ | CW NSD↓ | AA Avg.↑ | AA Worst↑ | AA NSD↓ |
> |:---:|:---:|:---:|:---:|:---:|:---:|:---:|:---:|:---:|:---:|:---:|:---:|:---:|
> | ZSKT | 38.79 | 24.30 | 0.308 | 37.74 | 19.80 | 0.350 | 32.04 | 21.50 | 0.256 | 10.65 | 1.20 | 0.783 |
> | CMI | 41.05 | 31.90 | 0.209 | 36.30 | 25.10 | 0.229 | 36.71 | 28.80 | 0.209 | 18.71 | 9.70 | 0.481 |
> | DeepInv | 47.28 | 33.50 | 0.202 | 35.20 | 23.00 | 0.230 | 42.14 | 28.10 | **0.198** | 16.71 | 5.60 | 0.531 |
> | Fast | 51.18 | 39.30 | 0.195 | 46.75 | 30.10 | 0.227 | 41.43 | 28.50 | 0.228 | 24.76 | 10.70 | 0.474 |
> | DFHL | 44.30 | 29.80 | 0.206 | 47.00 | 30.70 | 0.211 | 42.14 | 26.90 | 0.214 | 27.47 | 11.60 | 0.420 |
> | FERD(Ours) | **56.45** | **42.80** | **0.155** | **47.20** | **31.40** | **0.210** | **47.03** | **31.90** | 0.199 | **30.59** | **13.50** | **0.395** |
>
> As expected, the overall robustness decreases as $\epsilon$ increases. However, FERD consistently outperforms most baselines in terms of both worst-robustness and NSD across all $\epsilon$. This persistence of fairness improvement is attributed to our reweighting strategy and UTAEs. Even under stronger attacks, the reweighting strategy ensures sufficient training focus on weak-robust categories, while UTAEs force the student model to defend against diverse attack directions, avoiding overfitting to specific vulnerable categories.

---

> > ### Author Response · Authors · 2025-11-24
> >
> > **W3&Q2: Key parameters ($\tau$, $\gamma$, $\lambda$) are fixed without justification or sensitivity analysis, creating uncertainty about robustness across configurations.**
> >
> > A: We appreciate your suggestion. To address the concern about robustness across different configurations, we have conducted additional sensitivity analyses.
> >
> > **For the parameter $\tau$**, we supplement experiments under different $\tau$={0.3, 0.5, 0.7, 1.0}:
> > | $\tau$ | Clean Avg.↑ | Clean Worst↑ | Clean NSD↓ | FGSM Avg.↑ | FGSM Worst↑ | FGSM NSD↓ | PGD Avg.↑ | PGD Worst↑ | PGD NSD↓ |
> > | :---: | :---: | :---: | :---: | :---: | :---: | :---: | :---: | :---: | :---: |
> > | 0.3 | 79.17 | 62.10 | 0.116 | 61.28 | 42.20 | 0.186 | 55.83 | 35.10 | 0.215 |
> > | 0.5 | 79.26 | 65.90 | 0.110 | 61.04 | **46.10** | 0.160 | **54.93** | **39.80** | **0.197** |
> > | 0.7 | **80.21** | **66.70** | **0.100** | 61.63 | 44.20 | **0.155** | 54.24 | 35.40 | 0.200 |
> > | 1 | 80.14 | 63.20 | 0.112 | **61.98** | 45.80 | 0.163 | 54.65 | 35.30 | 0.220 |
> >
> > As shown in the table, the student achieves the optimal robustness and fairness under PGD attack when $\tau=0.5$. $\tau$ controls the sharpness of the reweighting distribution. When $\tau$ is too large, the distribution becomes too flat, failing to give enough attention to vulnerable classes. When $\tau$ is too small, the distribution becomes too sharp, causing the generator to overfit to a few specific classes. $\tau=0.5$ provides the best trade-off, effectively balancing the focus between vulnerable and robust categories.
> >
> > **For the parameter $\gamma$**, we have already discussed it in Section 5.3. We experiment with different $\gamma$={0, 0.1, 0.3, 0.5, 0.7, 0.9}. $\gamma$ balances the uniformity of attack targets against attack strength. We find that a low to medium $\gamma$ (e.g., 0.1 to 0.5) positively enhances the robustness and fairness of the student, while a high $\gamma$ (e.g., 0.7 or 0.9) reduces the attack strength of the adversarial examples, negatively affecting the robust performance. Therefore, we finally set $\gamma$ to 0.5.
> >
> > **For the parameter $\lambda$**, we have discussed it in Appendix A.7. We compare numerical and visual results under different $\lambda_{adv}, \lambda_{bn}, \lambda_{oh}, \lambda_{uni}$={1, 3, 5, 7}. We find that $\lambda_{adv}$ significantly improves the average robustness under all adversarial attacks, especially against FGSM and PGD. $\lambda_{bn}$ helps to improve the sample quality. $\lambda_{oh}$ controls the consistency of synthetic label. $\lambda_{uni}$ helps to alleviate the robust unfairness problem. In the context of DFRD, we prioritize robust fairness and practicality, which correspond to the function of $\lambda_{uni}$ and $\lambda_{bn}$, respectively. Therefore, we set them to 5 for more importance during training. Other terms work best at unit weight to maintain the primary classification capability without being overwhelmed by regularization terms.
> >
> > **W2: Experiments are confined to small datasets (CIFAR-10/100, Tiny-ImageNet).**
> >
> > A: Thanks for your suggestion. The ImageNet experiments require significantly longer training time. Since the review process is still ongoing, these runs are currently in progress. We will update the results as soon as the full ImageNet training completes.

---

> > > ### Author Response · Authors · 2025-12-03
> > >
> > > **W2: Experiments are confined to small datasets (CIFAR-10/100, Tiny-ImageNet).**
> > >
> > > A: We appreciate your suggestion. We add experiments on ImageNet with the XCiT-S12 as robust teacher and ResNet18 as student. We choose Fast as baselines, which is designed specially for large datasets like ImageNet. The results are as follows:
> > >
> > > | Method | Clean Avg.↑ | Clean NSD↓ | FGSM Avg.↑ | FGSM NSD↓ | PGD Avg.↑ | PGD NSD↓ |
> > > |:---:|:---:|:---:|:---:|:---:|:---:|:---:|
> > > | Fast | 25.82 | 0.699 | 3.44 | 1.740 | 3.07 | 1.823 |
> > > | FERD | **27.20** | **0.588** | **4.50** | **1.299** | **4.10** | **1.613** |
> > >
> > > As shown in the table, FERD outperforms baseline on ImageNet. This is attributed to the synthetic high-quality images and the fairness mechanism we designed, which leads to an improvement in both average robustness and robust fairness metrics.

---

### Author Response · Authors · 2025-12-03

Our paper investigates robust fairness in Data-Free Robustness Distillation(DFRD). We identify that this unfairness stems from equally distributed synthetic data and target-dependent vulnerabilities. We propose a framework FERD that involves a robustness-guided class reweighting strategy to focus on weak-robust classes, and introduce two techniques, Fair-Aware Examples (FAEs) and Uniform-Target Adversarial Examples (UTAEs), to force the model to develop a consistent and balanced defense against all attack targets. Extensive experiments show that FERD achieves state-of-the-art worst-class robustness under all adversarial attacks.

We sincerely thank the reviewers for their insightful feedback, which has significantly strengthened our paper and improved its readability. We have addressed all raised concerns and summarize the key revisions below:

- **Expansion of dataset, architecture, metric and baseline** (Reviewers wkEv, WX8r, VYHq, jJ8d): We have added experiments on ImageNet dataset, WRN-34-20 teacher model, two more robust fairness metrics and new DFRD baselines. The results further demonstrate the effectiveness and transferability of FERD framework.

- **Hyperparameter** (Reviewer wkEv): We have conducted additional sensitivity analyses for different parameters ($\tau$, $\gamma$, $\lambda$). We explain the role of the parameters and the reason for determining the parameter values. In addition, we test the robustness under different perturbation strengths $\epsilon$, and the results are still better than baselines.

- **Theoretical rigor and formalism** (Reviewers wkEv, WX8r): We have fully considered the suggestions of the reviewers and corrected the proof in the Appendix A.1 and A.2, which now include explicit assumption statements and formal proofs to support our framework. We have updated the manuscript with the modified contents highlighted in red.

- **Readability** (Reviewer WX8r): We have refined Section 4.3 to enhance expositional clarity. By streamlining the mathematical formalisms and condensing the presentation of loss functions, we have reduced notational density and improved the readability.

- **Reason for large average robustness gain** (Reviewer jJ8d): We have explained in detail to the reviewers that the reason for the large increase in average robustness is because of the high quality of our synthetic samples. FERD actively suppresses non-robust features related to adversarial perturbations. This mechanism produces high-quality, robust-fair samples, which significantly enhances the model's accuracy.

- **Adversarial examples generation** (Reviewer jJ8d): We clarified that the performance degradation of RSLAD attack generation stems from unreliable teacher predictions on low-quality synthetic samples. We contrasted this with FERD, demonstrating how our high-quality data generation ensures superior efficacy in adversarial distillation.

- **Effect in data-available distillation** (Reviewer jJ8d): We have transferred the fair components in FERD to data-available distillation and compared with SOTA methods in robust distillation, demonstrating the effectiveness of our framework. In addition, we explained the anomalously high PGD-20 accuracy is obtained using a prediction-based PGD protocol, which is different from what reviewers require, but this does not affect the final conclusion. We have re-measured the results as required and corrected them in the revision.

Overall, all concerns raised by Reviewers are addressed. Although the discussion phase finished before we could receive further feedback, we are confident that our comprehensive responses have fully resolved all concerns. We respectfully request that the Area Chair gives full consideration to the comprehensive responses and manuscript improvements detailed in this rebuttal.

---

### Meta-Review · Area_Chair_cMvs · 2026-01-04

**Summary:**

This paper received mixed reviews: one accept, two borderline accepts, and one borderline reject. The reviewers unanimously appreciated the novel and important problem setting (i.e., a combination of robust fairness and data-free robustness distillation). They also recognized the reasonable approach to the problem, extensive experiments, and favorable results. However, the reviewers also raised multiple crucial concerns with limited evaluation scope (wkEv, VYHq, jJ8d), lack of theoretical foundation (wkEv, WX8r), unclear contribution of the proposed techniques (jJ8d), insufficient quantitative analysis on fairness (WX8r), lack of analysis on the sensitivity to hyperparameters (wkEv), and some presentation issues (WX8r). The AC found that the authors thoroughly addressed most of these concern with thorough rebuttal and additional experiments during the author-reviewer discussion period. In particular, the rebuttal and revision provide more rigorous theoretical analysis (wkEv, WX8r), additional experiments with stronger and diverse attacks, with a different teacher model, and on a larger dataset (wkEv, VYHq), comparisons with diverse combinations of existing methods for robust fairness and robustness distillation (jJ8d), empirical analysis on the sensitivity to hyperparameters (wkEv), further evaluation with additional performance metrics (WX8r). The AC thus considers most of the critical concerns have been successfully addressed and recommends acceptance of the paper. The authors are strongly encouraged to add all the new results and discussions in the camera-ready version of the paper.

**Reviewer Concerns:**

The authors seem to have addressed most of the reviewers' major concerns successfully.
- Lack of theoretical foundation (wkEv, WX8r)
	- *The authors provided theoretical analysis in the revision. The reviewers pointed out this issue were originally positive.*
- Limited evaluation scope (wkEv, VYHq, jJ8d)
	- *The concerns raised by wkEv and VYHq are well addressed by additional experiments in the revision that have been conducted with stronger and diverse attacks, with a different teacher model, and on a larger dataset.*
	- *jJ8d asked evaluation under data-available adversarial distillation scenarios to better demonstrate the fairness improvement by the proposed method. The authors provided extensive experiments to address the request by the initial rebuttal and subsequent responses.*
- No analysis on the sensitivity to hyperparameters (wkEv)
	- *Addressed by additional experiments in the rebuttal.*
- Insufficient quantitative analysis on fairness (WX8r)
	- *Addressed by evaluations with two additional metrics in the revision.*
- Presentation issues (WX8r)
	- *The comment is mainly about overly large figures and too dense equations, and the AC found that they cannot be a ground for rejection.*
- Unclear contribution of the proposed techniques (jJ8d)
	- *The reviewer argued that the advantages of the proposed method combining robust fairness and robustness distillation over existing ones have not been well demonstrated, and asked further experimental analysis to better isolate the contribution of each component of the method (in particular, that for robust fairness). The authors have conducted numerous additional experiments with various baselines and in diverse settings to address the concerns, e.g., comparisons with combinations of existing (and latest) techniques for robust fairness and robustness distillation, evaluations under AutoAttack, and applications to diverse student network architectures. The reviewer pointed out that some of the new results look suspicious, but the authors found and fixed an error quickly and demonstrated that the conclusion remains correct even with updated results.*

**Reviewer Scores:**

Since the major concerns raised by the reviewers have been well assuaged, the AC expects that the positive reviewers would have stood by their original scores if the discussion phase has continued. Also, regarding the last comment of the negative reviewer in their initial review (i.e., "... concerns about baselines and context limit the current evaluation. I am open to revising my score based on the rebuttal.") and that the rebuttal and subsequent responses of the authors offer additional experiments with diverse baselines and extensive analysis on the contribution to robust fairness, the AC anticipates the reviewer would have upgraded their evaluation.

---

### Decision · Program_Chairs · 2026-01-26

Accept (Poster)